

# Hydrologic implications of projected changes in rain-on-snow melt for Great Lakes Basin watersheds

Daniel T. Myers[1], Darren L. Ficklin[1], and Scott M. Robeson[1]

[1]Department of Geography, Indiana University, Bloomington, Indiana 47405, USA

*Correspondence to*: Daniel T. Myers (danmyers901@gmail.com)

**Abstract.** Rain-on-snow (ROS) melt events reduce the amount of water stored in the snowpack while also exacerbating flooding. The hydrologic implications of changing ROS events in a warming climate, however, are still uncertain. This research used a calibrated and validated Soil and Water Assessment Tool (SWAT) hydrologic model, modified with energy budget equations to simulate ROS melt and forced with a climate model ensemble representing moderate greenhouse-gas

concentrations, to simulate changes to ROS melt in the North American Great Lakes Basin from 1960-2099. The changes to ROS events between the historic period (1960-1999) and mid-century (2040-2069) represent an approximately 30% reduction in melt in warmer, southern subbasins, but less than 5% reduction in melt in colder, northern subbasins. Additionally, proportionally more rainfall reduces the formation of snowpacks, with area-weighted winter+spring rain-to-snow ratios rising from approximately 1.5 historically to 2.0 by the end of the 21[st] century. Areas with historic mean winter+spring air

temperatures lower than -2 °C have ROS regimes that are resilient to 21[st] century warming projections, but ROS occurrence in areas that have mean winter+spring temperatures near the freezing point are sensitive to changing air temperatures. Also, relationships between changes in the timing of ROS melt and water yield endure throughout the spring but become weak by summer. As the influence of ROS melt events on hydrological systems is being altered in a changing climate, these conclusions are important to inform adaptive management of freshwater ecosystems and human uses in regions of the globe that are

sensitive to changes in ROS events.

## 1 Introduction

Rain-on-snow (ROS) melt events can have important implications for winter floods because of the combined impacts of rainwater and snowmelt runoff (Suriano and Leathers, 2018; Leathers et al., 1998). In places where ROS events are common, they have contributed to the majority of extreme floods, including locations in the United States Northwest, Upper Midwest,

Northeast, and Appalachians (Li et al., 2019). ROS events can occur across a wide swath of North America and Eurasia in areas that have substantial snowpack (Pomeroy et al., 2016; Rennert et al., 2009; Rössler et al., 2014; Sui and Koehler, 2001; Ye et al., 2008; Musselman et al., 2018), but their impact on hydrology extends beyond the cold season because snowpack conditions throughout the winter and spring influence the availability of groundwater and stream water later in the year (Blahušiaková et al., 2020; Jenicek et al., 2016; Myers et al., 2021b). Compared to thermally driven snowmelt rates, rainfall-



based melt events are often more short-lived and intense. As a result, ROS melt produces proportionally more runoff compared to temperature-based snowmelt, with lower rates of infiltration and groundwater recharge (Wilson et al., 1980; Earman et al., 2006). Thus, ROS melt events can lead to snow droughts and reduced water availability after the snow season because of the lost water storage (Harpold et al., 2017; Hatchett and McEvoy, 2018; Blahušiaková et al., 2020; Myers et al., 2021b).

In the North American Great Lakes Basin, ROS melt is associated with over 25% of the most extreme snowmelt
events (Suriano, 2020) and has been shown to influence hydrological droughts later in the year (Myers et al., 2021b). ROS events melt an average of 4 cm of snow per event in the Great Lakes Basin, but decreased in frequency by 37% from 1960-2009 (Suriano and Leathers, 2018). ROS melt typically occurs when a mid-latitude cyclone takes a more northerly track, transporting warm, moist air into the basin (Suriano, 2018). At the same time, the snow-water equivalent available in the snowpack is critical and average snow depths in the Basin decreased by 25% from 1960-2009 (Suriano et al., 2019). However,
the hydrological impacts of changing ROS melt amounts and frequencies in a transient climate are uncertain, as a decrease in snowpack could limit the amount of ROS melt, but also increase surface runoff from rain on bare ground during cold seasons.

This research combines outputs from an ensemble of downscaled climate models with a version of the Soil and Water Assessment Tool (SWAT) hydrologic model (Arnold et al., 1998) that incorporates a ROS melt modification (Myers et al., 2021b) to simulate climate change impacts to watersheds in the Great Lakes Basin. Our research asks, "How does ongoing
climate change alter ROS melt and hydrology in the Great Lakes Basin throughout the 21st century?" This research contributes to scientific knowledge by advancing our understanding of climate change impacts to watersheds, particularly concerning the impacts of ROS melt, improving our ability to manage water quantity and quality into the future. It is important to understand these climate change impacts to sustainably manage rivers and prepare for risks, both within the Great Lakes Basin and in ROS prone regions around the world.

## 2 Materials and methods

### 2.1 Study area

The North American Great Lakes Basin is the Earth's largest fresh surface water system (Environment Canada and USEPA, 1995), including portions of eight U.S. states and one Canadian province (Figure 1). To the north near Lake Superior, snow cover lasts an average of 180 days, but is as low as 107 days around Lake Erie (Suriano et al., 2019). The Great Lakes
Basin is experiencing a rapidly changing climate (Lehner et al., 2006; Environment Canada and USEPA, 1995), with annual average air temperatures having already risen nearly 1 °C since the early 20th century while annual total precipitation has risen approximately 10% (Wuebbles et al., 2019).



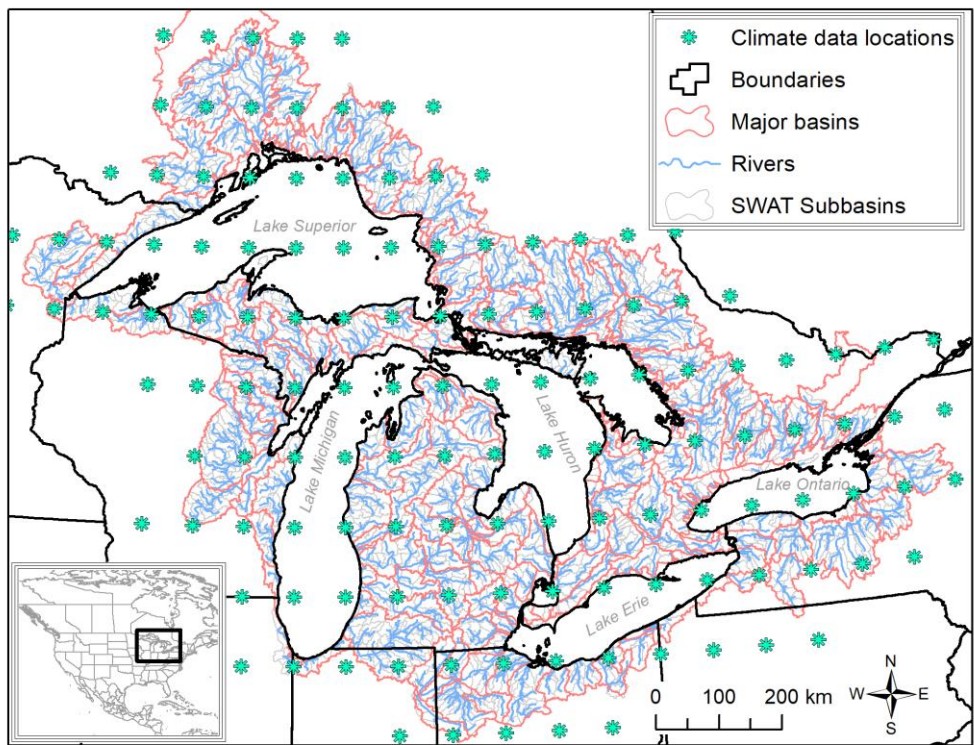

**Figure 1.** Study area map of the Great Lakes Basin, showing historical and projected climate data grid points and study river systems.


## 2.2 Hydrology simulation

We used the SWAT hydrological model (Arnold et al., 1998) with a modified snowmelt routine to simulate ROS melt. SWAT simulates hydrology with a water balance of inputs (precipitation), exports (evapotranspiration, surface runoff, groundwater flow, and lateral flow), and soil water storage. SWAT partitions precipitation into rainfall or snowfall based on
whether air temperature is above or below a temperature threshold (Fontaine et al., 2002). Groundwater flow was simulated with a shallow aquifer water balance and evapotranspiration was simulated using the Penman-Monteith method (Monteith, 1965; Ritchie, 1972).

A full description of the ROS modification, hydrology simulation, calibration, and evaluation can be found in Myers et al. (2021b). In short, the SWAT source code was modified to include an energy budget equation for snowmelt from the
SNOW-17 model (Anderson, 2006, 1973) that simulates ROS melt based on a function of air temperature, precipitation, wind, saturated vapor pressure, and atmospheric pressure. Previously, SWAT would simulate snowmelt using a snowpack temperature that was based on air temperature (Fontaine et al., 2002), which would not consider daily ROS melt events. Using a snowmelt module that can simulate ROS melt, such as our SWAT ROS model, reduces error and leads to more accurate





hydrological simulations in the Great Lakes Basin due to the more accurate simulation of the timing of snowmelt (Myers et
al., 2021b).

This study uses a calibrated version of the SWAT hydrological model for the Great Lakes Basin previously developed
in Myers et al. (2021b), based on historic climate inputs (Maurer et al., 2007). In Myers et al. (2021b), a sensitivity analysis
was performed using the PAWN method (Pianosi and Wagener, 2015) that identified 24 sensitive parameters. The model was
then calibrated at the daily time step with the A Multi-Algorithm Genetically Adaptive Multiobjective (AMALGAM)
algorithm (Vrugt and Robinson, 2007) using 99 stations for streamflow and 50 stations for snowpack snow water equivalent
(SWE). SWE was estimated from the gridded North American snow depth dataset (Mote et al., 2018) using a function of snow
depth, precipitation, temperature, and time of year (Hill et al., 2019). Nash Sutcliffe Efficiency (NSE; Nash and Sutcliffe,
1970) and the revised Index of Agreement ($d_r$; Willmott et al., 2012) were the objective functions for streamflow, while mean
absolute error (MAE; Willmott and Matsuura, 2005) was the objective function for snowpack SWE. The SWAT ROS model
for the Great Lakes Basin simulated historic streamflow at the daily time step with an NSE of 0.38 (with 29% of stations
greater than 0.5) and a $d_r$ of 0.62 (Myers et al., 2021b). The model simulated historic snowpack SWE at the daily time step
with an MAE of 26 mm. Calibrated parameters for this model can be found in Table S1 in Myers et al. (2021b).

## 2.3 Climate projections

The calibrated hydrological model was forced with 1950-2099 climate projections from downscaled and bias-
corrected outputs of the Coupled Model Intercomparison Project Phase 5 (CMIP5) multi-model ensemble (Taylor et al., 2012;
US Bureau of Reclamation, 2013; Maurer et al., 2007). These models were downscaled to a 1° latitude/longitude grid (Figure
1) and bias-corrected using the Bias-Correction Constructed Analogues method (Maurer et al., 2010), which corrects bias by
quantile mapping with historic data (US Bureau of Reclamation, 2013). One-degree grid resolution was chosen for these
projections because it matched the resolution of our snowpack data for calibration (Mote et al., 2018). It was important to have
our snow data and climate projections at the same resolution so that we could calibrate the model for snow processes at the
same scale as the response to air temperature and precipitation. These processes could be sensitive to differences between grids
(Rajulapati et al., 2021; Winchell et al., 2013; Myers et al., 2021a).

Global climate models (GCMs) can be a major source of uncertainty when modeling the hydrological impacts of
climate change (Wang et al., 2020; Chegwidden et al., 2019). Thus, nineteen climate models for the Representative
Concentration Pathway (RCP) 4.5 were used to account for variation in climate projections (Table 1). RCP 4.5 is a moderate
greenhouse-gas scenario that considers long-term changes in emissions, land cover change, the global economy, and climate
change mitigation (Thomson et al., 2011). The mean of this multi-model ensemble was used to represent our projection
(Christensen et al., 2010), with the standard deviation of the GCM ensemble shown in Figure 2. Simulations of ROS changes
generally agree across CMIP5 RCP's (4.5 and 8.5) until mid-century, but then diverge by late-century (Musselman et al.,
2021a).



**Table 1.** Climate models for the RCP 4.5 scenario used in the research. Full names for each modeling center can be found in Table S1.

| Climate Model | Modeling Center | Country | Citation |
|---|---|---|---|
| ACCESS1-0 | CSIRO-BOM | Australia | (Bi et al., 2013) |
| BCC-CSM1-1 | BCC | China | (Wu et al., 2014) |
| CanESM2 | CCCMA | Canada | (Arora et al., 2011) |
| CCSM4 | NCAR | USA | (Gent et al., 2011) |
| CESM1-BGC | NSF-DOE-NCAR | USA | (Long et al., 2013) |
| CNRM-CM5 | CNRM-CERFACS | France | (Voldoire et al., 2013) |
| CSIRO-Mk3-6-0 | CSIRO-QCCCE | Australia | (Rotstayn et al., 2012) |
| GFDL-ESM2G | NOAA GFDL | USA | (Dunne et al., 2012) |
| GFDL-ESM2M | NOAA GFDL | USA | (Dunne et al., 2012) |
| INM-CM4 | INM | Russia | (Volodin et al., 2010) |
| IPSL-CM5A-LR | IPSL | France | (Hourdin et al., 2013) |
| IPSL-CM5A-MR | IPSL | France | (Dufresne et al., 2013) |
| MIROC-ESM | MIROC | Japan | (Watanabe et al., 2011) |
| MIROC-ESM-CHEM | MIROC | Japan | (Watanabe et al., 2011) |
| MIROC5 | MIROC | Japan | (Watanabe et al., 2010) |
| MPI-ESM-LR | MPI-M | Germany | (Giorgetta et al., 2013) |
| MPI-ESM-MR | MPI-M | Germany | (Giorgetta et al., 2013) |
| MRI-CGCM3 | MRI | Japan | (Yukimoto et al., 2012) |
| NorESM1-M | NCC | Norway | (Bentsen et al., 2013) |

## 2.4 Analyses

Hydrological outputs from the SWAT model were aggregated to the boundaries of regulatory river basins for the United States (Hydrologic Unit Code HUC 8; USGS, 2022) and Canada (Tertiary-Level watersheds; Government of Ontario, 2022) using spatial averaging. Aggregating our subbasins into the regulatory major river basins from the USGS HUC 8 and Ontario Tertiary Level watersheds data allowed us to compare SWAT outputs with the existing basin structure and facilitates comparisons and discussions of our results with other studies.

Results were analyzed by comparing averages and extreme high events among historic (1960-1999), mid-21st century (2040-2069), and late-21st century (2070-2099) time periods at the subbasin scale, based on water years (1 October to 30 September). The mid-21st century period was the focus for informing water resources management and because of better





agreement among the models. Calculations of basin-wide averages were weighted by subbasin area. Seasons were defined as

winter (December, January, February), spring (March, April, May), summer (June, July, August), and fall (September, October, November). ROS melt events were defined as days with >1 mm rainfall on >1 mm snowpack SWE (Jeong and Sushama, 2018). The ROS center of volume statistic, defined as the day of the water year when half the total volume of ROS melt has passed, was used to examine changes to the timing of ROS melt events during a water year, adapted from Hodgkins et al. (2003).

Extreme hydrological events were identified as the 0.95 quantile daily event for high flows. Pearson's correlation was used to evaluate the strength of linear relationships, with "significant" relationships defined as $p < 0.05$. Boxplots and percentiles were spatially weighted by major river basin area (Willmott et al., 2007). Finally, the SWAT model outputs for water yield represent the area-averaged water export through the outlet in mm.

## 3 Results

### 3.1 Precipitation and air temperature projections

Across the Great Lakes Basin, CMIP5 ensemble average annual precipitation and air temperatures are projected to increase between the historic 1960-1999 period and mid-21st century using RCP 4.5. Spatially averaged annual precipitation rises from 839±63 mm (mean and standard deviation of GCM ensemble) during 1960-1999 to 892±77 mm by the mid-21st century (a 6.3% increase), while spatially averaged annual air temperatures rise from 5.3±0.7 °C during the 1960-1999 period

to 7.9±1.0 °C (Figure 2a and b). Changes in ensemble mean winter+spring air temperatures are most prominent in northern parts of the Basin, where winter+spring air temperatures are projected to rise approximately 3 °C between the historic 1960-1999 period and mid-21st century using the RCP 4.5 scenario (Figure 3a and b). Using the GCM ensemble mean, changes in mean winter+spring rainfall are strongest in the Lake Superior region, where the amount of rainfall is projected to increase over 40% (Figure 3c and d; Figure 2d). Northern areas experience the least change in ensemble mean winter+spring snowfall

amount between the historic 1960-1999 period and mid-21st century, while southern parts of the Basin have a decrease in snowfall over 10% (Figure 3e and f; Figure 2e). Further, our model shows that winter+spring rain to snow ratios over the basin (calculated by dividing the total winter+spring rainfall by total winter+spring snowfall) rise from around 1.5 historically to 2.0 by end of century, which means that proportionally more rainfall could contribute to the declines in snowmelt and snowpack SWE (Figure 2c).




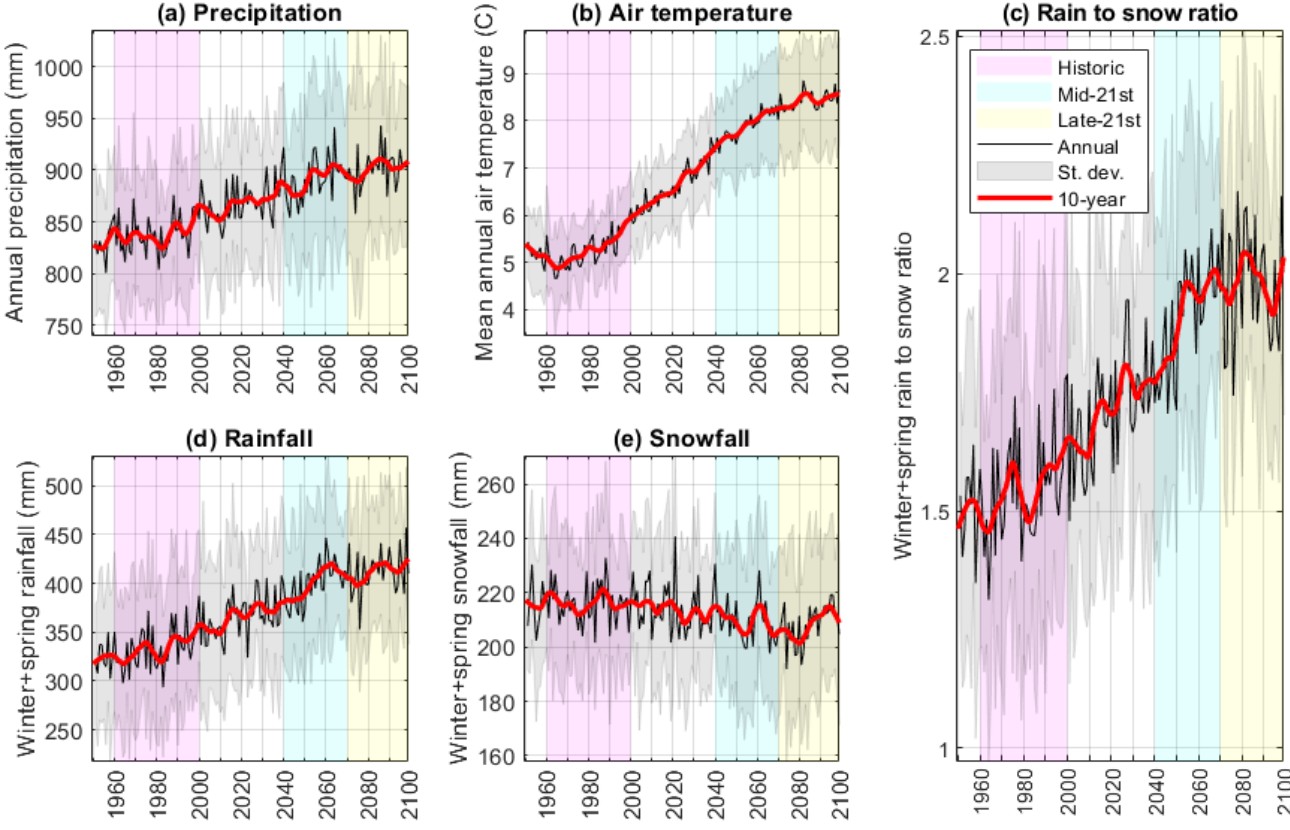

**Figure 2.** Basinwide ensemble-average a) annual total precipitation, b) annual air temperature, c) winter+spring rain to snow ratio, d) winter+spring rainfall, and e) winter+spring snowfall from the climate input data 1950-2099, with 10-year averages (red lines), based on the RCP 4.5 pathway. Shading indicates historic 1960-1999 (red), mid-21st century 2040-2069 (blue), and late-21st century 2070-2099 (yellow) periods, as well as ensemble standard deviations (grey).





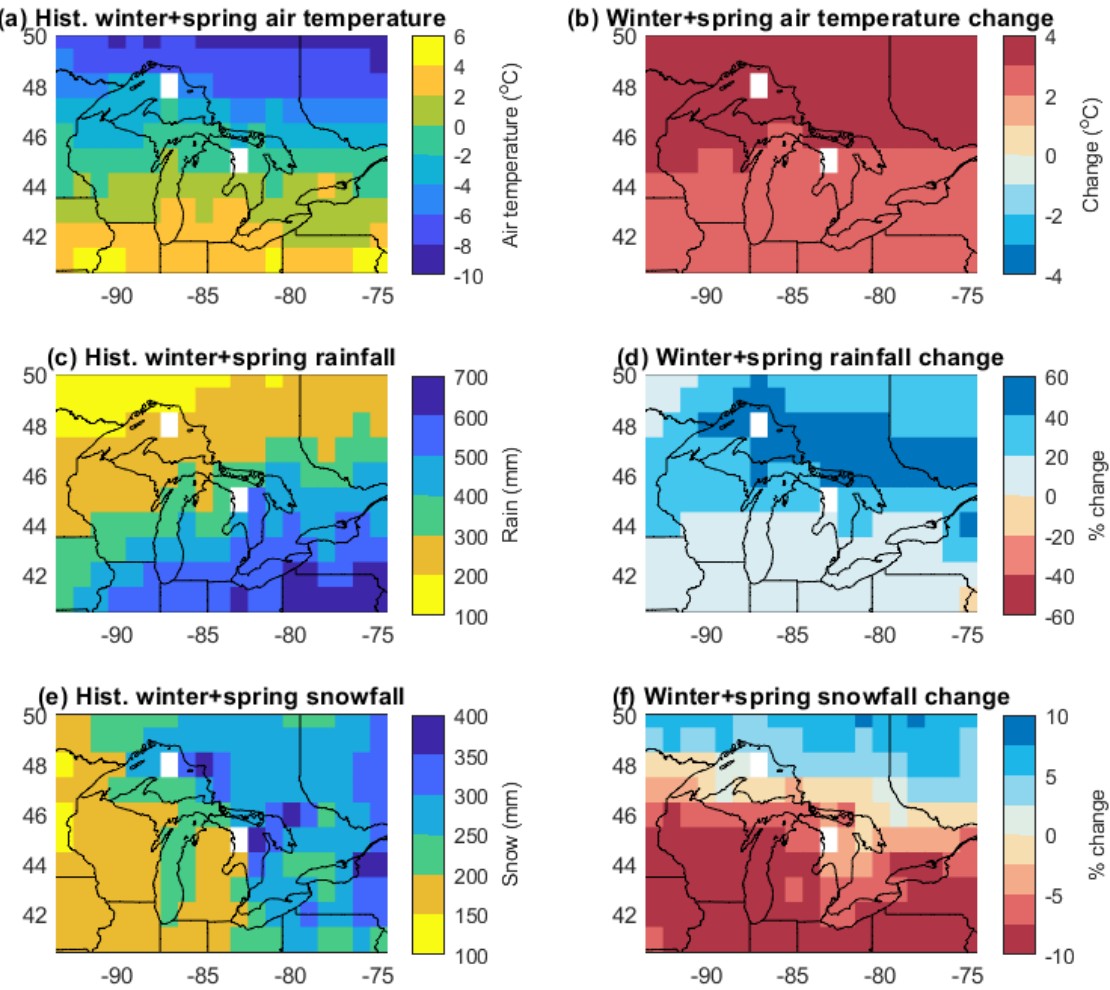

**Figure 3.** Historic and projected climate changes for the Great Lakes Basin between the historic 1960-1999 period and mid-21st century (2040-2069) for RCP 4.5 ensemble mean winter+spring: (a and b) air temperatures, (c and d) rainfall, and (e and f) snowfall, based on water years. Left panels show historic amounts while right panels show absolute or percent changes.

## 3.2 Snowpack and snowmelt projections

Between the historic 1960-1999 period and mid-21$^{st}$ century, winter months generally see an increase in the amount of ROS melt for individual subbasins due to the increased amount of rainfall (not snowfall) under RCP 4.5 projections. For instance, the area-weighted median (value of a ranked set where at least half the total area is ranked lower; Willmott et al., 2007) amount of February ROS melt among subbasins was 7.0 mm historically, but rises to 10.5 mm by mid-21$^{st}$ century, an increase of 50% (Figure 4a). Similarly, area-weighted median January ROS melt rises from 3.2 mm historically to 5.1 mm in the mid-21$^{st}$ century. In the spring, the amount of ROS melt decreases due to the reduction in snowpack. For instance, the area-





weighted median amount of April ROS melt among subbasins was 52.3 mm historically but falls to 22.9 mm by the mid-21st century, a decrease of 56%.

Mean monthly snowmelt (including temperature-based melt and ROS melt), among individual Great Lakes Basin subbasins, is projected to experience a drastic decrease and shift to earlier timing in the spring by the mid-21st century (Figure 4b) using the RCP 4.5 pathway. Historically the maximum snowmelt overall has been in April with an area-weighted median of 85.3 mm, while the March median snowmelt was 44.8 mm. By mid-21st century the median amount of snowmelt among subbasins is still 44.8 mm in March but only 39.5 mm in April, which is a 54% April decrease between the two periods.

Changes in the amount of monthly snowmelt among individual subbasins are affected by changes to ROS melt
amounts because days that have ROS melt occurrences account for greater than 50% of total snowmelt for most subbasins from December through April (Figure 4c). Temperature-based snowmelt is usually a slower process, while ROS melt events combined with temperature-based melt on these days can rapidly melt snowpack. However, the proportion of melt occurring during December ROS days decreases from an area-weighted median of 71% historically (1960-1999) to 59% by mid-21st century (a decrease of 12%). With warmer temperatures, temperature-based melt can have more of an influence on total
snowmelt. The proportion of total annual snowmelt from ROS tends to increase in the northern and eastern parts of the Great Lakes Basin, but decrease in the south and west, between the historic 1960-1999 period and mid-21st century, by about 5% in each direction as temperatures warm (Figure 5a and b). Additionally, snowpack SWE decreases throughout the winter and spring. For instance, by March in the mid-21st century, only 61.6 mm of area-weighted median snowpack SWE is left in the Basin, compared with a median of 104.0 mm historically (a decrease of 41%) (Figure 4d).



**Figure 4.** Changes in RCP 4.5 ensemble mean monthly a) rain-on-snow (ROS) melt, b) total snowmelt (including temperature-based melt and rain-on-snow), c) the proportion of monthly snowmelt from rain-on-snow events, and d) snowpack snow water equivalent (SWE) for 158 individual major river basins of the Great Lakes Basin between the historic 1960-1999 and mid-21st century periods. Boxplots display the median and interquartile range of results for major river basins and are weighted by river basin area (Willmott et al., 2007).





**Figure 5.** Changes in (a and b) proportion of total snowmelt from rain-on-snow, (c and d) subbasin ROS melt, and (e and f) frequency of ROS events between the historic 1960-1999 period and mid-21st century. Left panels indicate historic conditions, while the right panels indicate absolute or percent changes. Projections represent the RCP 4.5 ensemble mean.






### 3.3 Rain-on-snow melt projections

ROS melt is affected by the changing climate and there will be different intensities and frequencies of ROS melt events in the future using the RCP 4.5 pathway. Northernmost subbasins near Lake Superior experience the least changes in the annual amount of ROS melt between the historic 1960-1999 period and mid-21[st] century, with less than a 5% change.

However, the central and southern areas of the Basin experience large decreases in annual ROS melt, with the greatest reduction in southern subbasins in Michigan and southern Ontario with a >30% decrease in the amount of annual ROS melt, and a >20% decrease in the frequency of ROS events (Figure 5c-f). Overall, the ensemble average amount of annual snowmelt during ROS events, at the major river basin scale using RCP 4.5 models, changes by -42% to +1%, with a basinwide area-weighted average of -22%. Meanwhile, the range among climate projections in the ensemble for this change in basinwide average annual

snowmelt during ROS events is -50% to -3%, suggesting that the climate models agree that there will be a reduction in annual ROS melt for the basin overall.

Additionally, northern and central subbasins around eastern Lake Huron and the southern shore of Lake Superior tend to see a slight increase in the annual frequency of ROS events of +5%, while more southern subbasins experience the greatest decrease in frequency around -25% (Figure 5e and f). This is because the northern subbasins, which maintain substantial

snowpack throughout the winter and spring and have temperatures well below freezing, have ROS frequencies that are resilient to increases in air temperature (through mid-century), while ROS frequencies in southern subbasins with winter+spring air temperatures around the freezing and melting points are sensitive to even small perturbations in air temperature, with threshold-like responses around these temperature points to the partitioning between rainfall and snowfall.

Following the earlier timing of ROS melt, the center of volume for ROS melt (the day of the water year when half of

the total annual ROS melt is passed) decreases between the historic 1960-1999 period and mid-21[st] century. Historically, the ROS melt center of volume ranged from day 145 (23 February) in the southern part of the Great Lakes Basin to day 207 (26 April) in the northern part (Figure 6a). By the 2050s, the ROS melt center of volume becomes earlier and ranges from day 134 (12 February) to day 198 (16 April), which is approximately two weeks earlier (Figure 6b).





**Figure 6.** Changes in (a and b) the center of volume (COV) for rain-on-snow melt (ROS) events (the day of the water year [DOY] when half the total annual rain-on-snow melt is passed), (c and d) daily mean winter+spring snowpack SWE, and (e and f) high (0.95 quantile) daily winter+spring water yields between the historic 1960-1999 period and mid-21st century. Left panels indicate historic conditions, while right panels indicate percent changes. Projections represent the RCP 4.5 ensemble mean. The water year lasts from 1 October to 30 September.



### 3.4 Relationships with climate and snowpack

The cause of the reduction in annual ROS melt across the Basin is largely from a reduction in snowpack SWE due to the rising air temperatures. Although an increase in ensemble average winter+spring precipitation in major river basins by +7% to +15% contributes to ROS melt (Figure 2a, Figure 3c-f), its influence is negated by a decrease in the annual amount of winter snowpack SWE in river basins of -10% to -52% (Figure 6c and d). This is due to proportionally more winter rainfall, as the winter+spring rain to snow ratio increases from 1.55±0.32 (mean and standard deviation of GCM ensemble) during the historic 1960-1999 period to 1.91±0.31 by mid-21$^{st}$ century (Figure 2c).

Changes in the annual amount of ROS melt are strongly correlated with historic winter+spring snowpack SWE (r=0.87, p<0.001) and also with the frequency of ROS events (r=0.82, p<0.001). These relationships are also related to location, with higher latitudes experiencing less of a change in ROS melt and frequency (Figure 7a and b). A decrease in winter+spring SWE is consistently larger in the southern subbasins, where warming mean temperatures above the freezing and melting points reduce the ability for snow to accumulate (Figure 6d). The lack of snowpack means that ROS melt events may be unable to occur as often or as intensely in these southern subbasins as they were able to historically. Changes to the amount of annual ROS melt and frequency of ROS events are not correlated with historic winter and spring total precipitation amounts (Figure 7c and d), as the type of precipitation is more influential, and depends on air temperatures (and thus latitude).



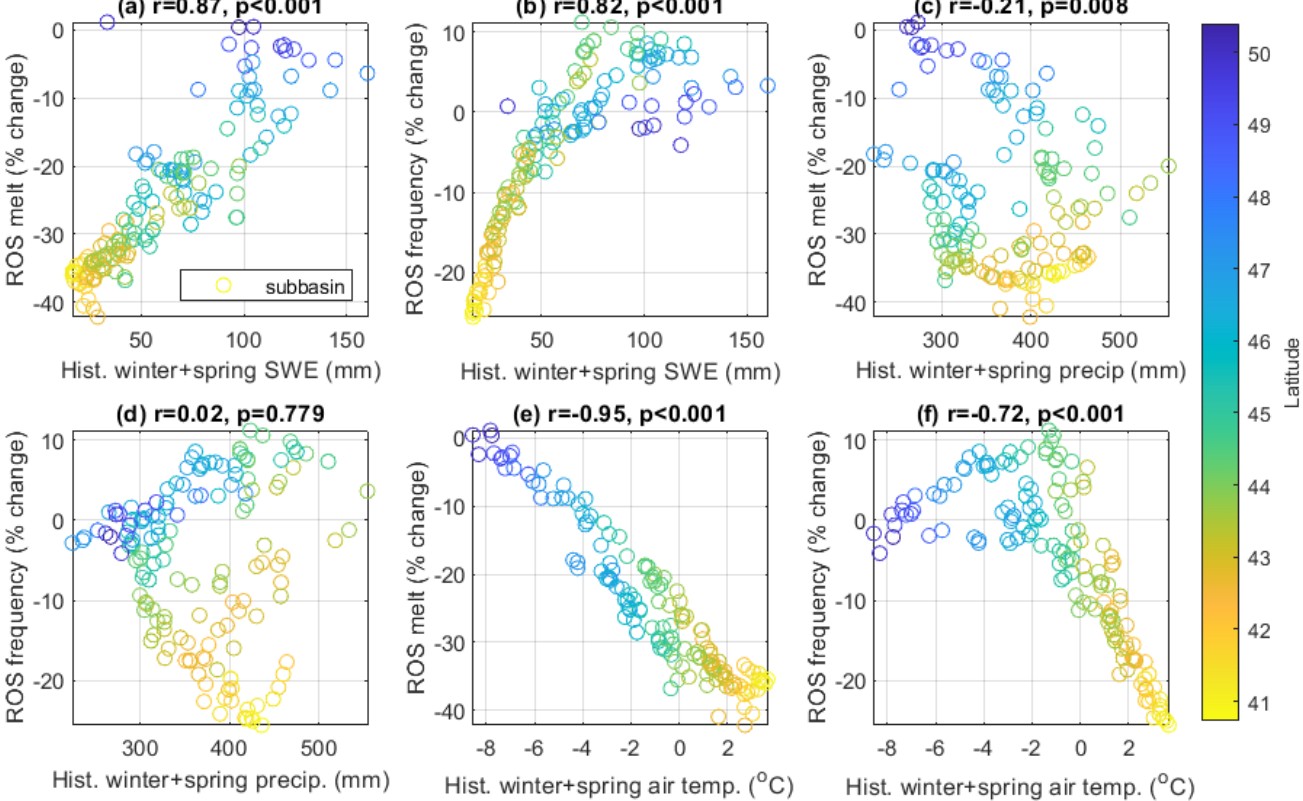

**Figure 7.** Relationships between the historic 1960-1999 conditions and the percent change from the historic period to mid-21st century for RCP 4.5 ensemble means. a) historic winter+spring snow water equivalent (SWE) and rain-on-snow (ROS) melt amount change, b) historic winter+spring SWE and ROS frequency change, c) historic winter+spring precipitation and ROS melt change, d) historic winter+spring precipitation and ROS frequency change, e) historic winter+spring air temperature and ROS melt change, and f) historic winter+spring air temperature and ROS frequency change. Colors are coded with the latitudinal gradient.

Mean historic winter and spring air temperatures have a strong relationship with changes to ROS melt. Subbasins that have colder winter+spring air temperatures during the historic 1960-1999 period have weaker changes to the amount of ROS melt (r=-0.95, p<0.001; Figure 7e) and frequency of ROS events (r=-0.72, p<0.001; Figure 7f). For example, subbasins in the Lake Superior watershed historically have mean winter+spring air temperatures of around -5 °C (Figure 3a). Even with an increase in mean winter+spring air temperatures of +3 °C (Figure 3b), temperatures remain cold enough to have a reduced influence on ROS occurrences for much of the winter and spring. Areas where ROS is most sensitive to the changing climate are the central and southern river basins of the Great Lakes Basin, where historic mean winter+spring air temperatures were around 0 °C. Here, perturbations to air temperatures due to climate change can have the greatest effects on ROS melt, leading to decreases in the amount of annual ROS melt stronger than -30%, as the locations that experience these temperatures are often near the freezing and melting points (Figure 5c and d).





## 3.5 Relationships with seasonal hydrology

For extreme events, most of the basin (92% of subbasins) experiences a decrease in the magnitude of daily extreme high (0.95 quantile) winter+spring water yields (an area-weighted average of -9%) between the historical 1960-1999 period and mid-21$^{st}$ century under RCP 4.5, although northern subbasins have less change (Figure 6e and f). The reduction of snowpack SWE means that the size of high spring snowmelt flows can be diminished, even as cold season precipitation increases. For extreme water yields in spring months alone (March through May), 93% of major river basins experience a decrease in the magnitude of daily extreme high water yields due to climate changes, averaging an area-weighted -14% decrease between the 1960-1999 period and mid-21$^{st}$ century (Fig. S1). However, for winter months alone (December through February), there is an increase in the magnitude of daily extreme high water yields for 84% of subbasins, averaging an area-weighted +17% increase. The reason for this distinction is that extreme spring snowmelt events move earlier in the year and become smaller, while extreme winter melt events become more common. This suggests that climate change is thus affecting the timing of extreme water yield events to be earlier in the year.

Changes in extreme winter+spring high water yields for major river basins are positively correlated with changes in annual ROS melt amount (r=0.56, p<0.001) but not frequency (r=0.11, p=0.169), as the frequency of ROS melt events would not imply the size of the events. Extreme winter+spring high water yields are also positively correlated with changes in winter+spring precipitation amount (r=0.66, p<0.001). Extreme winter water yields alone (December through February) are positively correlated with changes in annual ROS melt amount (r=0.59, p<0.001) and frequency (r=0.82, p<0.001), as well as winter precipitation amount (r=0.37, p<0.001). This suggests that, even though high spring snowmelt floods can become diminished due to lack of snowpack, a combination of ROS melt and increased winter precipitation increases the severity of high winter (December through February) flows, particularly as more winter precipitation falls as rain, and the timing of these extreme events becomes earlier in the year (Figure 2c). This is supported by the findings of greater winter snowmelt and ROS melt, as well as reduced spring snowmelt, on a monthly basis between the historic 1960-1999 period and mid-21$^{st}$ century (Figure 4a and b).

The earlier center of volume for ROS melt has a lagged response on monthly water yields for the major river basins that lasts throughout the spring but becomes obscured by summer. There is a positive correlation between changes in the ROS melt center of volume and March (r=0.42 (p<0.001) and April (r=0.70, p<0.001) water yields between the historic 1960-1999 period and mid-21$^{st}$ century, due to the influence of ROS melt on daily water yields when snowpacks are actively melting (Figure 8a and b). In May, this relationship abruptly switches to negative (r=-0.33, p<0.001), as the reduced impact of ROS means that less water is rapidly exported from the watershed during large melt events, and there is a delayed contribution to water yield (Figure 8c). However, by the summer months of June, July, and August, correlations are weak if any (Figure 8d-f). Changes in summer water yields have much stronger relationships with summer precipitation than they do with ROS, for instance in July (r=0.74, p<0.001) and August (r=0.84, p<0.001), obscuring the influence of the timing of ROS melt on summer

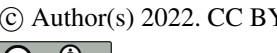



water yields over time. This suggests that the earlier timing of ROS melt events (by center of volume) is related to hydrology through the spring, although the relationship can be obscured by summer by other factors such as summer precipitation.

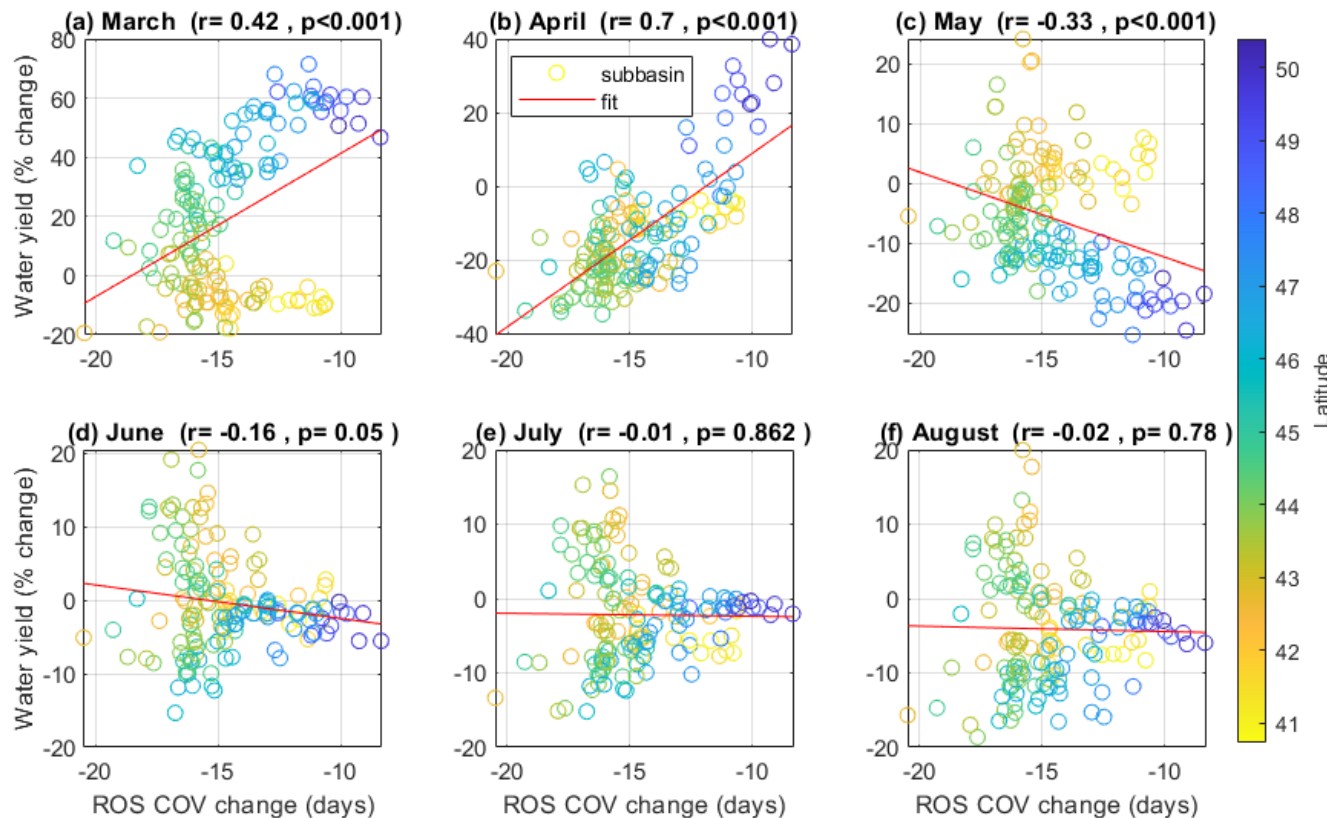

**Figure 8.** Correlations of the change in the center of volume (COV) for rain-on-snow melt events between the historic 1960-1999 period and mid-21st century, with the percent change in monthly water yield during that time, for spring and summer months. Markers represent projections of the RCP 4.5 ensemble mean for individual major river basins. Colors are coded with the latitudinal gradient. The water year lasts from 1 October to 30 September.

## 4 Discussion

It is important to understand the factors affecting spatial variability of ROS changes in the Great Lakes Basin to realize the impacts of this variability on aquatic resources. Spatial variability in ROS melt changes has previously been shown to occur because of differences in latitude, elevation, and atmospheric processes (Pan et al., 2018; Ye et al., 2008; Jeong and Sushama, 2018; Cohen et al., 2015), although there were not large elevation changes in the Great Lakes Basin to observe elevation-based variability at our scale. There are particularly large increases in ROS melt runoff predicted for northeastern North America, but decreases in more southern latitudes due to a decrease of snow cover (Jeong and Sushama, 2018). The





frequency of ROS melt events can be affected by latitude because of its association with air temperature, precipitation type (rain or snow), and snow cover (Suriano, 2022). This aligns with our findings for the Great Lakes Basin that the change in ROS melt amounts decreases by mid-century and is strongly related to latitude, with the greatest decreases in southern subbasins where snowpack becomes exhausted, as latitude affects whether mean winter air temperatures will be around the threshold around the freezing and melting points where ROS is most sensitive to changes in climate.

Similarly, an understanding of the temporal factors affecting variability in ROS changes can provide insights to the timing of hydrological impacts. Temporally, the frequency of ROS events in a warming climate has the potential to increase as more rain falls on snowpack, but decline after a warming threshold is reached and the snowpack becomes scarce (Beniston and Stoffel, 2016). For instance, in eastern Russia, the frequency of ROS events has historically increased with warming air temperatures because of more winter rainfall, at a rate of 0.5 to 2.5 events per °C air temperature increase, but future increases could be limited by a lack of snow in warmer regions (Ye et al., 2008). Prior research by Suriano (2022) has found a slightly increasing trend in the annual frequency of ROS melt events in the Great Lakes Basin from 1960-2009, at the rate of approximately 0.0 to 0.1 events per year, although there was a decrease in spring (March and April) ROS melt events as large as -0.03 events per year. We compared historic 1960-1999 and mid-21st century (2040-2069) time periods and found a general decrease in the frequency of ROS melt events over a longer time period. It is possible that ROS melt events increase in frequency at first as more cold-season rain falls on snowpack, but then decrease once that snowpack is substantially reduced by the mid-21st century. It is also possible that the difference was due to variation in how ROS events were defined in each study. Suriano (2022) also found snowfall amounts to be a dominant control on the frequency of North American ROS melt events.

The earlier timing of ROS melt (and earlier passage of its annual center of volume) has the potential to influence other parts of the hydrologic cycle. In the eastern United States, since 1940, the timing of the center of streamflow volume passing through gages in the winter and spring has become earlier at a rate of 1.6 days per decade, due to increasing air temperatures in snowmelt-driven regions and earlier snowmelt occurrences (Dudley et al., 2017). A similar trend of earlier snowmelt timing has been found for the western United States as well (Stewart et al., 2004; Musselman et al., 2021b). The earlier timing of snowmelt aligns with what our projections show for the Basin overall, due to winter precipitation increases and ROS, supporting that the earlier center of volume for ROS melt could influence spring water yields. Also, following a 2 °C warming scenario, prior research has found that the frequency of ROS melt events are expected to become approximately one month earlier in the eastern United States as cold-season snowfall switches to rainfall (Li et al., 2019), which aligns with our findings for the earlier center of volume for ROS melt over the water year. As the contributions of snowmelt to peak spring water yields become weaker due to the earlier timing of ROS events, vegetation green-up can have a more dominant influence on spring hydrographs in a changing climate (Khodaee et al., 2022).

Changing rain to snow ratios can have meaningful impacts to hydrological systems. The rain to snow ratio is important because it was previously found to be the primary avenue for changing air temperatures to affect snowpack in the Sierra Nevada of California, USA, exacerbating runoff during early-season flooding (Huang et al., 2018). As more precipitation falls as rain





rather than snow, the size of floods from rainfall and ROS events can far exceed the size of typical snowmelt-driven floods, due to the rapid contributions of rainfall and ROS runoff, with the largest increases being over 2.5 times in size for the western United States (Davenport et al., 2020). The rain to snow ratio can also influence the size and timing of spring snowmelt and summer baseflow (Huntington et al., 2004). Thus, it could help explain the earlier center of volume of ROS melt for the Great

Lakes Basin by mid-21st century.

        The proportion of total snowmelt from ROS or temperature-based melt also has important hydrological impacts. Research in the western United States has found that climate change can decrease the speed at which snowpack melts, as warmer air temperatures mean that there will be bare ground later in the snowmelt season while radiative energy fluxes are high, and more snowmelt occurs during the colder part of winter when energy fluxes are low, causing snowmelt to be a slower

process (Musselman et al., 2017). This increase in the proportion of temperature-based melt to total snowmelt reflects a transition to slower, earlier snowmelt, and helps explain the decrease in high spring streamflow that historically influenced hydrological regimes in the Great Lakes Basin (Hodgkins et al., 2007). The decrease in the proportion of total snowmelt from ROS in the Great Lakes Basin could also contribute to groundwater recharge (Earman et al., 2006; Wilson et al., 1980) and the increases in May water yields in subbasins where the annual amount of ROS decreases (Figure 8c), as the water would not

have been rapidly exported from the subbasins in earlier ROS events.

**5 Conclusions**

        Climate change is disrupting rain-on-snow patterns globally, potentially impacting ecosystems, communities, and economies in regions where these events are prevalent. This study used the Soil and Water Assessment Tool (SWAT) rain-on-snow melt (ROS) model, which builds upon SWAT by incorporating energy budget equations to simulate ROS melt (Myers

et al., 2021b), to study the impacts of climate change on ROS melt due to altered snowpack, air temperatures, and precipitation. An ensemble of RCP 4.5 climate projections (representing moderate greenhouse gas concentrations) was used to study relationships. Although winter+spring precipitation increases in the Great Lakes Basin by the mid-21st century, compared with historic 1960-1999 amounts, its influence on ROS melt is limited by an exhausted snowpack with warmer air temperatures, particularly for southern subbasins. Winter+spring rain to snow ratios from the climate input data rise from around 1.5

historically to 2.0 by end of the 21st century, so proportionally more rainfall decreases snowpack SWE. Changes in ROS melt are positively correlated with snowpack snow water equivalent and winter+spring precipitation.

        We find that relationships with ROS patterns and latitude are strong in the Great Lakes Basin, with northern subbasins having air temperatures that remain well below freezing most of the winter+spring and are resilient to temperature increases, while southern subbasins that had mean winter+spring temperatures around freezing historically are more sensitive to changes

in air temperatures. The changing temperature directly affects whether snowpack would form or melt, or whether precipitation would be snow or rain. We expect this result of increased sensitivity for ROS changes to apply to cold regions around the globe with average winter+spring air temperatures around 0 °C. With increasing air temperatures, temperature-based snowmelt



can have more of an influence on total monthly snowmelt, as the proportion of monthly melt from ROS decreases (e.g. -12% in December) between the historic 1960-1999 period and mid-21$^{st}$ century.

We also find there are temporal relationships with ROS melt timing in the Great Lakes Basin by mid-21$^{st}$ century, as the center of volume (the day of the water year when at least half the total ROS melt volume has passed) becomes earlier by approximately two weeks compared to the historic 1960-1999 period. The temporal scale of impacts from this earlier timing on monthly water yields lasts through the spring (positively correlated in March and April, but negative in May), although these relationships can be obscured by summer because of changing summer precipitation. Also, extreme winter (0.95 quantile)

daily water yields increase in magnitude +17% between the historic 1960-1999 period and mid-21$^{st}$ century, but extreme spring daily water yields decrease -14% as high snowmelt floods become diminished, showing how climate change can affect the timing of extreme hydrological events.

    Finally, it is important that future work involve collaborations outside the academic realm so that the findings of climate change impacts to ROS melt can inform management of aquatic resources (Meadow and Owen, 2021) and engage

communities with the research (Serreze et al., 2021). Implications of this work, specifically involving the influence of changing ROS melt on extreme hydrological events and future water availability, as well as the climate-related sensitivities to changing ROS melt, could help prepare the management of ecosystems and human water uses for the climatic changes of the 21$^{st}$ century and beyond.

## Supplementary material

Supplementary material for this article is available online for Table S1 and Fig. S1.

## Funding

This work was supported by the Indiana University Geography Department, William R. Black Fellowship, Indiana University Sustainability Research Development Grant, National Science Foundation [grant numbers DBI-1564806 and CNS-0521433], Indiana University Pervasive Technology Institute, Lilly Endowment, Inc., Indiana METACyt Initiative, and Shared

University Research Grants from IBM, Inc. to Indiana University. Any opinions, findings, and conclusions or recommendations expressed are those of the authors and do not necessarily reflect the views of the National Science Foundation.

## Data availability

The data and SWAT ROS model used in this study are publicly available from Mendeley Data at

http://dx.doi.org/10.17632/bfypd4wpcn.1.

## Competing interests

The authors declare that they have no conflict of interest.



**Acknowledgements**

We thank Ram Neupane, Alejandra Botero-Acosta, and Dan Li for guidance with this study. We also thank Indiana
University's University Information Technology Services High Performance Computing team for technical support. We
acknowledge the World Climate Research Program's Working Group on Coupled Modelling, which is responsible for CMIP,
and we thank the climate modeling groups (listed in Table S1 of this paper) for producing and making available their model
output. For CMIP the U.S. Department of Energy's Program for Climate Model Diagnosis and Intercomparison provides
coordinating support and led development of software infrastructure in partnership with the Global Organization for Earth
System Science Portals.

**Author contribution**

Daniel T. Myers: Conceptualization, Methodology, Software, Validation, Formal Analysis, investigation, Data Curation,
Writing – Original Draft, Writing – Review & Editing, Visualization. Darren L. Ficklin: Conceptualization, Methodology,
Software, Resources, Writing – Original Draft, Writing – Review & Editing, Supervision, Project Administration, Funding
Acquisition. Scott M. Robeson: Conceptualization, Methodology, Software, Resources, Writing – Original Draft, Writing –
Review & Editing, Visualization.

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
