# Peer review of "Table S1. Modeling center and institution names. Adapted from CMIP5 Model Groups and their Terms of Use (<https://pcmdi.llnl.gov/mips/cmip5/availability.html>)"

_EGUsphere, 2022_

## Author Comment (AC4)

**Reviewer 1**
General comments:

This study is a novel investigation that is of interest to the professional community and in-line with the aims and scope of the journal. The topic is appropriately introduced with justification provided for the specific objectives. While some additional details on the statistical testing could be added (see below), the methodological approach appears logical and reproduceable. The results are organized around specific themes with figures that enhance understanding and are aligned with the final conclusions. Prior to supporting acceptance and publication, there are a small number of outstanding concerns with the manuscript that are addressed below as specific comments.

> Response: We thank the reviewer for the thorough and helpful review that has improved the quality of the manuscript.

Specific comments:

The proportions of historical ROS melt [to total melt] is larger here than a variety of previous findings for the region. For instance, Welty and Zeng (2021) find extreme ROS occurrence is approximately 24% for the Great Lakes basin, similar to the value the authors give on line 34 at over 25% of extreme ablation events being ROS. Looking at all ROS events, not just extreme, the maximum value to date I am aware of for this region is found in Suriano (2022). This notes between 30-50% of ablation is ROS in the eastern lakes, compared to less than 20% in the extreme northern/western regions. While the results here have a similar spatial pattern to Suriano (2022), with more ROS in the eastern lakes and less to the north and west, the magnitudes are rather different. Given one of the primary results of this study is the detection of large decreases in ROS events under the RCP4.5 scenario relative to historical period, it is warranted to provide further discussion on the robustness of the historical model values relative to observations. This appears absent from the manuscript currently and should be incorporated into the discussion section of the revision.

> Response: For our study, we used the definition of Jeong and Sushama (2018) to define an ROS event, as this definition was being used by them to project future climate impacts using RCP's across North America, and was based on the ROS definitions of studies before them. Thus, we defined an ROS event as >1 mm rainfall on >1 mm SWE and snowmelt occurring, so our results would be directly comparable with theirs. We now include an additional Discussion section (4.2) and figure that discusses the comparability of findings of historic ROS melt with other studies, and objectively evaluates our models against historic observed data, to be added at the location of page 19, line 351 of the preprint.
>
> "*Previous work by Jeong and Sushama (2018), whose definition of ROS we adopted, has found comparable estimates of historic frequencies of ROS events as we did,*

*approximately 10-20 ROS days per year in the Great Lakes Basin. Also, Jeong and Sushama (2018) report an historic average annual amount of ROS runoff of approximately 100 mm or greater throughout the Basin, which is of a similar magnitude to our historic estimates. Jeong and Sushama evaluated their models using historic observations and found that spatial patterns in ROS were captured reasonably well, though errors could arise from uncertainties in the data driving their models rather than problems. Nonetheless, other studies have used different definitions of ROS events and/or have reported variable findings for the Great Lakes Basin. For instance, Welty and Zeng (2021) defined an ROS occurrence as air temperature >0 °C and precipitation >5 mm during 2-day extreme snowmelt events (e.g. >50 mm and the top 10 events over a 30 year historic period), which includes far fewer ROS events that we did. Additionally, Suriano (2022) defines an ROS event as a snow depth decrease of at least 1 cm with average daily temperature >0 °C, at least 0.01 cm precipitation, and no more than 2.54 cm snowfall (by depth) the previous day, over a 1960-2009 historic period. With this definition, Suriano (2022) reports a historic frequency of approximately 5 to 15 ROS events per year in the Great Lakes Basin.*

*To objectively verify the robustness of our historic estimates, we identified ROS amounts and frequencies in observed data using the same approach and definition as our GCM-forced SWAT model. The historic climate observations were from Maurer et al. (2007), used in Myers et al. (2021b), and our historic SWE observations were from Myers et al. (2021b), which had been estimated off the daily gridded North American snow depth dataset (Mote et al., 2018), both in a matching 1° latitude/longitude grid with 50 evaluation points over the Great Lakes Basin. We found that for historic annual estimates of ROS melt, the mean among the gridded evaluation points for our GCM ensemble was 120 mm, while the mean calculated from observations was 118 mm, which was not a significant difference (p=0.90). For individual evaluation points, the estimates of annual ROS melt were positively related with an MAE of 33 mm (Fig. S2a). This suggests that our GCM ensemble was reasonably estimating historic ROS melt amounts in the Basin. We also found that the historic observations estimated a mean average annual ROS frequency of 20 days across the evaluation points, which was greater than the mean of 12 days estimated by our GCM ensemble for the points over our historic 1960-1999 period (p<0.001). This was because our ROS definition included historic observed events that were the result of natural stochasticity in snowpack SWE amounts (i.e., sporadic daily increases or decreases in the SWE data, rather than "clean" modeled melt). Thus, our definition overestimated the frequency of ROS days when applied to historic observations, compared with our modeled ROS frequency, due to the additional stochastic small melt events identified by the criteria, with an MAE of 8 days (Fig. S2b). However, when ROS amounts are accumulated over the season, this issue is remedied (Fig. S2a)." The smoothing produced by the observed data being aggregated to a 1 degree grid could also affect the comparisons between the historic observed and modeled estimates.*

[Figure]

**Figure S2.** For the 50 gridded climate and snowpack evaluation points in the Great Lakes Basin, a) Comparison of historic (1960-1999) mean annual ROS melt amounts calculated for observed data with those modeled by our ensemble of climate projections, and b) The same comparison for the mean annual frequency of ROS events.

References:

Jeong, D. Il and Sushama, L.: Rain-on-snow events over North America based on two Canadian regional climate models, Clim. Dyn., 50, 303–316, https://doi.org/10.1007/s00382-017-3609-x, 2018

Suriano, Z. J.: North American rain-on-snow ablation climatology, Clim. Res., 87, 133–145, https://doi.org/10.3354/CR01687, 2022.

Welty, Josh, and Xubin Zeng. "Characteristics and causes of extreme snowmelt over the conterminous United States." Bulletin of the American Meteorological Society 102.8 (2021): E1526-E1542.

The authors acknowledge on line 126 the threshold used for statistical significance for their correlation tests. However, it is unclear if any significance testing was conducted for the rest of the study. Was any sort of t-test or difference of means testing conducted for the results comparing the historical period to the mid-century period? If not, this should be considered by the authors to aid in differentiating meaningful changes from ones still within the noise.

Response: We now have included significance testing for our comparisons of ROS and climate between the historic and mid-21st century periods. We also now describe the

approach for this in the methods, and provide instances where significance testing is used below. However, throughout our revision we keep effect size as the focus, rather than statistical significance, following the guidance of previous work (Wasserstein et al., 2019; Ziliak & McCloskey, 2008)

"*For comparisons between time periods, significance was tested by comparing annual area-weighted ensemble-average values for the Great Lakes Basin between the historic (1960-1999, n=40 years) and mid-21st century (2040-2069, n=30 years) periods using two-tailed unpaired t-tests.*" (to be added at page 6, line 126 of the preprint)

"*Spatially averaged annual precipitation increases 6.3% from 839±63 mm (mean and standard deviation of GCM ensemble) during 1960-1999 to 892±77 mm by the mid-21st century (p<0.001), while spatially averaged annual air temperatures increase from 5.2±0.7 °C during the 1960-1999 period to 7.9±1.0 °C (a 2.7 °C increase, p<0.001).*" (to be updated at page 6, line 132 of the preprint)

"*Further, our model shows that winter+spring rain to snow ratios over the basin (calculated by dividing the total winter+spring rainfall by total winter+spring snowfall) increase from around 1.5 historically to 1.9 by mid-century (p<0.001), which means that proportionally more rainfall could contribute to the declines in snowmelt and snowpack SWE.*" (to be updated at page 6, line 141 of the preprint)

"*Overall, the ensemble average amount of annual snowmelt during ROS events, at the major river basin scale using RCP 4.5 models, changes by -42% to +1%, with a basinwide area-weighted average of -22% (p<0.001).*" (to be updated at page 12, line 197 of the preprint)

References:

Wasserstein, R. L., Schirm, A. L., & Lazar, N. A. (2019). Moving to a world beyond "p< 0.05". The American Statistician, 73(sup1), 1-19.

Ziliak, S., & McCloskey, D. N. (2008). The cult of statistical significance: How the standard error costs us jobs, justice, and lives. University of Michigan Press.

---

## Author Comment (AC5)

**Reviewer 2**

A representative simulation of ROS melt events is important for improving hydrological modeling practice in snow dominated region. It is valuable to look into the future impact of ROS melt events under climate change. This is exactly what this work intends to address. However, the current manuscript is not yet ready for publication, due to two points :

> Response: We thank the reviewer for the thoughtful feedback which we have used to improve the quality and clarity of the manuscript.

1. This work utilized a calibrated SWAT ROS model to simulate the hydrological process using CMIP5 climate projections. All analyses are based on the assumption that this calibrated model is representative. However, as described "The SWAT ROS model for the Great Lakes Basin simulated historic streamflow at the daily time step with an NSE of 0.38 (with 29% of stations greater than 0.5) and a dr of 0.62 (Myers et al., 2021b). The model simulated historic snowpack SWE at the daily time step with an MAE of 26 mm", the model cannot be well considered well-calibrated with a low NSE of 0.38 for discharge simulation. Moreover, 26 mm MAE for daily SWE is a considerable high bias in comparison to the SWE value of the study area (e.g. Figure 4). The median SWE value of many months is around 50 mm or lower. GCM climate projections are highly uncertain already. A hydrological model with high bias will make the combination much worse. As a consequence, it is not reasonable to trust the analyses of this work about future climate change impact, even the analysis strategy is comprehensive. Therefore, the authors should implement the climate change investigation based on a reasonably well-calibrated SWAT ROS model. Moreover, detailed information about the rationality of the calibrated SWAT model is necessary but missing. Such information should be properly added to this paper or its supplementary material for its readers. The authors simply cited the paper that developed and evaluated the SWAT ROS model (reference below). But it is not open-access.

Myers, D. T., Ficklin, D. L., and Robeson, S. M.: Incorporating rain-on-snow into the SWAT model results in more accurate simulations of hydrologic extremes, J. Hydrol., 603, 126972, https://doi.org/10.1016/J.JHYDROL.2021.126972, 2021b.

> Response: Our multi-station calibration of streamflow and snowpack with a single global parameter set across the hydrologically diverse Great Lakes Basin enabled us to represent and compare streamflow and snowpack processes more comprehensively and objectively, as we describe further in the text below. This approach allowed us to verify that any spatial variation in ROS was actually due to variation in climate forcings, rather than artifacts of model parameters being overfitted to individual watersheds, and facilitated the communication of model performance in different hydrologic systems. Thus, as the evaluation statistics (e.g., NSE) are arbitrary when comparing different modeling approaches, we believe that the benefits of this approach outweigh the sacrifice in NSE in comparison with more regional or station-selective calibration options, which could have higher NSE values but be overfit to individual systems and produce more uncertainty

whether climate forcings or different model parameters would be causing the spatial ROS variation.

We attempted several new calibrations of the Great Lakes Basin SWAT ROS model aimed to improve model performance, including expanding parameter ranges within reason and removing the least sensitive parameters from the calibration. However, we were unable to improve model performance for simulating streamflow and snowpack beyond that of the Myers et al. 2021b model, which had been heavily experimented on with different calibration strategies and evaluations during that study. We believe that the accuracy of our model is actually good considering the uncertainties of all the spatially aggregated datasets that go into it (climate, snowpack, soils, etc.), and especially when considering the use of our daily time step, which incorporates high temporal scrutiny in our evaluations over a large geographic area. Previous work by Kalin et al. (2010) has stated that arbitrary interpretations of performance metrics for models at small temporal scales should be relaxed compared to what would be expected for models at coarse (e.g., monthly) time steps, for instance that an NSE between 0.3 and 0.5 could fit criteria for satisfactory model performance.

We also expanded our description of model evaluations to include more synopses from the Myers et al. 2021b study, and a figure which depicts the geographic dispersion of stations that calibrated well and those that did not calibrate as well. Figure 2 (below) now shows how stations that perform well for streamflow and snowpack simulation are dispersed throughout the Great Lakes Basin. We apologize that the Myers et al. 2021b paper is not open access. We are happy to share this paper, through the editor if you prefer. In addition, we provide more background for our choice of the Myers et al. 2021b model to represent spatial variability in ROS across the Great Lakes Basin using a single global parameter set and including evaluation stations that perform well along with those that do not perform as well. We provide the updated text below, to be added at the location of page 4, line 84 in the preprint.

*"The SWAT ROS model for the Great Lakes Basin simulated historic streamflow at the daily time step with an average NSE of 0.38 (with 29% of stations greater than 0.5, 48% of stations having NSE > 0.4, and a maximum NSE of 0.71) and a $d_r$ of 0.62 (Myers et al., 2021b). The model simulated historic snowpack SWE at the daily time step with an MAE of 26 mm (Figure 2a-c). Calibrated parameters for this model can be found in Table S1 in Myers et al. (2021b). We also investigated seasonal model performance for only days when ROS melt was occurring, and found that the SWAT ROS model we use had an MAE of 8.6 mm, 9.4 mm, and 5.8 mm, respectively for simulating melt on those days in the winter, spring, and fall.*

[Figure]

**Figure 2.** Evaluations for simulating historic streamflow Nash Sutcliffe Efficiency (NSE), streamflow revised Index of Agreement (d_r), and snowpack mean absolute error (MAE) at the daily time step from Myers et al., (2021b).

*The multi-station evaluation we used with a global parameter set (from Myers et al., 2021b) enabled us to represent streamflow and snow processes more comprehensively and comparatively across the hydrologically diverse basin. Thus, any spatial differences in ROS would represent actual differences due to spatial variation in climate forcings, rather than artifacts of regionally calibrated parameter sets which could simulate processes differently. Alternatively, a regional calibration may have improved model performance for some stations, but at the expense of no longer being able to objectively evaluate spatial variation in ROS across the basin. We included stations that performed well (NSE > 0.5) along with stations that did not perform as well in our evaluations across the basin. Along with providing transparency about model performance in different hydrologic systems, this allowed us to verify that our model was performing as well as possible for simulating many watersheds and hydrological processes in a comparative way, and openly communicate instances where the model did not perform as well (Figure 2). Alternatively, we could have used fewer stations and evaluated the model to a more geographically limited system, which would have resulted in better performance for simulating those stations, but at the expense of misrepresenting the diversity of hydrological responses (e.g., snowpack) across the basin. We believe that the benefits of our approach, to be representative of variable hydrological responses and use a global parameter set, in providing confidence to our interpretations of spatial ROS variation outweigh the sacrifice of model performance for some stations."*

For background from the Myers et al. 2021b publication, the SWAT ROS model was an improvement over the traditional SWAT model (not including ROS) which had an average daily streamflow NSE of -0.05 and average snowpack NSE of 48 mm). Also, the SWAT ROS model we used in this study was more accurate for simulating historic snowpack SWE at all 50 calibration stations than the traditional SWAT model. The traditional SWAT had larger snowmelt simulation errors of 10.4 mm, 10.2 mm, and 6.4 mm, respectively, for ROS days in winter, spring, and fall. In Myers et al. (2021b), the SWAT ROS model for the Great Lakes Basin was further evaluated in comparison with the traditional SWAT model for streamflow and snowpack simulation over 504 randomly generated parameter sets. The SWAT ROS model we used in this study improved performance (in comparison to the traditional SWAT) for simulating daily streamflow in

99.4% of the potential parameter sets, and for simulating daily snowpack in 100% of the sets. This demonstrated the nearly universal benefits for model performance of including ROS in simulations of Great Lakes Basin hydrology as we did.

In response to other reviewer comments, we are also adding a section 4.2, which discusses our historic ROS estimates in comparison with other studies, and is described in our public comment to Reviewer 1. In this section, we also objectively evaluate the historic ROS melt amount and ROS frequency estimates of our GCM ensemble against estimates from historic observations. This showed that our SWAT ROS model with GCM forcings was reasonably simulating ROS melt in the Great Lakes Basin in comparison with historic evidence.

Kalin, Latif, et al. "Predicting water quality in unmonitored watersheds using artificial neural networks." Journal of Environmental Quality 39.4 (2010): 1429-1440.

2. Future climate projects have large uncertainty. When evaluating climate change impacts, it is more reasonable to discuss the trend or relative changes rather than absolute quantities. The authors should shorten such contents and keep the necessary ones only. Besides, Figure 2 shows different behaviors of climate driving force during different future periods. It would be interesting to investigate the corresponding hydrological signatures of different future periods. Although, as described in section 2.4, the analyses of future period include mid-21st century and late-21st century. Throughout the paper, the result and analysis of late-21st century is almost none. Please complete such missing parts.

Response: We now primarily focus on relative changes in our analyses, while some absolute changes are also mentioned to provide context for the magnitude of the changes. An example to be added at the location of page 8, line 157 of the preprint is below.

"*For instance, the area-weighted median (value of a ranked set where half the total area is ranked lower; Willmott et al., 2007) amount of January ROS melt among subbasins increases 59% from 3.2 mm historically to 5.1 mm by mid-21st century (Figure 5a). Similarly, area-weighted median February ROS melt rises 50% from 7.0 mm historically to 10.5 mm in the mid-21st century. In the spring, the amount of ROS melt decreases due to the reduction in snowpack.*"

For consistency and clarity in our findings, we decided to remove the isolated references to late-21[st] century from the paper, and focus on mid-21[st] century changes only. We focus on mid-21[st] century because it is the period that the models (and RCPs) generally agree about climate changes, and because we expect that it will be the most meaningful focus for water resource managers in the Great Lakes Basin. Thus, we updated the preprint's Figure 2 (now Figure 3) to end after mid-21[st] century and clarified our text to alleviate any confusion about future time period.

[Figure]

**Figure 3.** Basinwide ensemble-average a) annual total precipitation, b) annual air temperature, c) winter+spring rain to snow ratio, d) winter+spring rainfall, and e) winter+spring snowfall from the climate input data, with 10-year averages (red lines), based on the RCP 4.5 pathway. Shading indicates historic 1960-1999 (red) and mid-21[st] century 2040-2069 (blue) periods, as well as ensemble standard deviations (grey).

---

## Author Comment (AC6)

**Community Comment 1**

This review was prepared as part of graduate program Earth & Environment (course Integrated Topics in Earth & Environment) at Wageningen University, and has been produced under supervision of dr Ryan Teuling. The review has been posted because of its potential usefulness to the authors and editor. Although it has the format of a regular review as was requested by the course, this review was not solicited by the journal, and should be seen as a regular comment. We leave it up to the author's and editor which points will be addressed.

Title paper: "Hydrologic implications of projected changes in rain-on-snow melt for Great Lakes Basin watersheds"

Overall impression

Rain on snow (ROS) melt events can have a big influence on their surroundings and can be either big or small. This manuscript investigates the impact of the climate change on the ROS events in the Great Lakes Basin in northern United States and few states of Canada for the period of 1960 to 2099, focussing on the period of 2040-2069 by the use of the model Soil and Water Assessment Tool (SWAT) with the addition of an energy budget equation to project the ROS events. With the results they looked at relationships and correlations between the different obtained variables. In general, the ROS events tend to happen earlier in the year by mid-21st century compared to the historical 1960-1999 values. The rain to snow ratio changes from around 1.5 historically to 2.0 at the end of the 21st century. This all also has influence on the water yield of the basins which also shift to earlier in the year.

The paper is nicely written and is important in these times of changing climate. The text has a good structure as the results are divided in understandable blocks. The variables are also captured in some good figures, although some changes should be made. Based on these comments, I would recommend publication with minor revisions.

> Response: We thank you, members of the Integrated Topics in Earth & Environment course, for the valuable feedback. We have incorporated the comments into our manuscript and present our responses. We will also include you in the acknowledgements of our revision.

General comments,

Firstly, as definition of a rain-on-snow event in the analysis section is stated that an event occurs on days with >1 mm rainfall on >1 mm snowpack SWE. By reading the Jeong and Sushama, 2018 paper this definition is not complete. It should be days with >1 mm rainfall on >1 mm snowpack SWE and decreasing SWE. By using the wrong definition rain-on-snow events could be wrongfully depicted in the results, and thus possible differences in conclusions. This should be solved by correctly using the definition and changing this in the paper.

Response: Thank you for pointing out our error in stating how we defined ROS events. You are correct that the definition must include snowmelt occurring, which we included in our script but hadn't mentioned in the text. We have now updated it to "ROS melt events were defined as days with >1 mm rainfall on >1 mm snowpack SWE and snowmelt occurring (Jeong and Sushama, 2018)." (to be updated at page 6, line 121 of the preprint)

Secondly, in the research question is stated that the change of rain-on-snow melt and hydrology due to climate change will be assessed for the 21st century. In the method an argument is made about that for informing water resources management and because of better agreement with the models the primary focus will be on the mid-21st century (2040-2069). In the rest of the paper late-21st century is only mentioned for the change in ratio of area-weighted winter+spring rain-to-snow. Which has only a 0.1 change to the ratio change of the mid-21st century where the mid-21st century has a better agreement to the models and thus will probably be more accurate. Also in the conclusion at the end is stated "could help prepare ….. for the climatic changes of the 21st century and beyond" where nothing is known for this latter time period with results from this paper. Thus, either the research question has to be rewritten to only the mid-21st century together with the conclusion or the late-21st century should be included in the rest of the analysis.

Response: Thank you for pointing out this need for consistency. We have removed references beyond the mid-21st century from the research question and conclusions, as well as other locations where late-21st century had been mentioned in the text, to focus our study consistently on mid-21st century changes.

"*Our research asks, "How does ongoing climate change alter ROS melt and hydrology in the Great Lakes Basin by the mid-21st century?"*" (to be updated at page 2, line 44 of the preprint)

"*Implications of this work, specifically involving the influence of changing ROS melt on extreme hydrological events and future water availability, as well as the climate-related sensitivities to changing ROS melt, could help prepare the management of ecosystems and human water uses for the climatic changes of the mid-21st century.*" (to be updated at page 20, line 380 of the preprint)

Lastly, again a comment on the research question but now because change in hydrology due to climate change is asked in the question. When stating hydrology I expect more variables to be analysed then only water yield, the rest of the analysed variables either belong to the change in climatic values such as rain and temperature or change in rain-on-snow events. Also, in the methods groundwater is mentioned to be modelled (line 65) but later not analysed in the results. So, either the phrasing of the research question should be altered or more hydrological variables should be assessed in the paper. This is important as the goal of this paper is to inform water managements to prepare for the changes due to climate change. For them the groundwater or

runoff variables are also very important, and as they change with changing ROS melt (as said in line 39-41) this should be addressed.

> Response: Our results focused on water yield because it is normalized streamflow, which our model was calibrated and evaluated for. Similarly, we report snowpack that our model was calibrated and evaluated for. Other variables such as groundwater and soil water clearly are important, but we provided limited results for them, because of space or data limitations and because they had not been specifically calibrated and evaluated for using the observed data. This is definitely an area of future research that we hope to look into further. Thus, we added a sentence about the importance of this additional research at page 20, line 380 of the preprint.

> "*Future work could also investigate how changing ROS conditions affect other components of the water balance including groundwater and soil water storage in the Great Lakes Basin.*"

Comments,

1.  In lines 312 -320, Myers mentions the difference in findings with Surianon mentioning that the studied times differ. But as the simulation used in this study is done for 1960-2099, the same time periods could be compared as the data will be present after simulations. Why not analyse the same time period (1960-2009) as Suriano to make this comparison possible, to diminish this suggestive difference.

    > Response: We decided to remove this comparison since it was not definitive, and we now provide more concrete details about historic ROS in a new Discussion section 4.2 (described in our public comment to Reviewer 1). We used 1960-1999 as the historic time period for this study because that matches previous work we have performed and evaluated (Myers et al. 2021b) using our SWAT ROS model, and fits with the time period of the historical climate observations we used (Maurer et al. 2007).

    > Maurer, E. P., Brekke, L., Pruitt, T., and Duffy, P. B.: Fine-resolution climate projections enhance regional climate change impact studies, Eos, Trans. Am. Geophys. Union, 88, 504, https://doi.org/10.1029/2007eo470006, 2007.

    > Myers, D. T., Ficklin, D. L., and Robeson, S. M.: Incorporating rain-on-snow into the SWAT model results in more accurate simulations of hydrologic extremes, J. Hydrol., 603, 126972, https://doi.org/10.1016/J.JHYDROL.2021.126972, 2021b.

2.  Lines 338-340, the speculation of influence of rain-to-snow ratio on size and timing of spring snowmelt and summer baseflow is made. This could be analysed by simply calculating the

correlation between the COV (center of volume) and the rain-to-snow ratio which are variables present in the results.

Response: We updated the Discussion text with our results for this relationship at the location of page 19, line 339 of the preprint. We decided instead of the correlation to state the size of basin-average effects.

"*Thus, the rain to snow ratio could help explain the earlier COV of ROS melt for the Great Lakes Basin by mid-21st century, since we found that as the basin-average rain to snow ratio increases from approximately 1.5 historically to 1.9 by mid-21st century, the COV of ROS melt becomes two weeks earlier.*"

3. Line 99, "thus. Nineteen climate models …… were used". There is a missing argument about why you use 19 models instead of more or less. Please add an argument.

Response: To improve clarity in this choice, we added the following at the location of page 4, line 100 of the preprint:

"*We chose to include nineteen climate models because that was the total number of RCP 4.5 models that had been downscaled and bias corrected in the multi-model ensemble (Maurer et al., 2007).*"

Maurer, E. P., Brekke, L., Pruitt, T., and Duffy, P. B.: Fine-resolution climate projections enhance regional climate change impact studies, Eos, Trans. Am. Geophys. Union, 88, 504, https://doi.org/10.1029/2007eo470006, 2007.

4. Line 165, the word "drastic" is used. Drastic is not a quantitative value as it is more an subjective use when not supported with arguments. Either rewrite the sentence or add an argument.

Response: We removed the word to avoid the subjectivity.

5. Lines 103-105, this information is a good/ better argument for the statement in line 118-119. As it now seems as random added information but potentially better used in the latter argument.

Response: We moved the text as suggested.

6. Line 167 , it is as a reader unclear whether the value mentioned for March is also the maximum or the value at the time of the maximum of April. This should be clarified.

Response: We reworded this sentence for clarity at the location of page 9, line 166 of the preprint:

*"Historically the maximum snowmelt overall has been in April with an area-weighted median of 85.3 mm, while the March median snowmelt has been less at 44.8 mm. By mid-21st century the median amount of monthly snowmelt among subbasins reaches a maximum at 44.8 mm in March, but drops to only 39.5 mm in April, which is a 54% April decrease between the two periods."*

7. Line 172, it is unclear if "proportion of melt" means temperature based melt or ROS based melt. This should be clarified.

   Response: We added an explanation to the location of page 9, line 172 of the preprint to be clearer:

   *"However, the proportion of melt occurring during December ROS days (compared with all December melt) decreases from an area-weighted median of 71% historically (1960-1999) to 59% by mid-21st century (a decrease of 12%)."*

8. Comment on figures in general: in the captions some abbreviations are written out where others are only posted as abbreviation. For the consistency of the paper this should be the same in all figures.

   Response: We fixed the figure captions so that the abbreviations are described at the first mention, and then abbreviated each time after.

9. Line 212, why is 2050s mentioned here instead of the same formulation used in the rest of the paper: "mid-21$^{st}$ century"? This could confuse the reader.

   Response: We changed it to "mid-21$^{st}$ century" to avoid the confusion.

10. Figure 6, Instead of "high flows" state "high water yields" in figure title of 6 e and f as stated in the caption for clarification.

    Response: We updated the figure as suggested, but for space wrote "high WYLD" in the figure title then defined WYLD as water yield in the caption.

11. Figure 7, could be better depicted when the figures with frequencies (b, d and f) are on the right and with melt (a, c and e) on the left.

    Response: We updated the figure as suggested.

Specific comments,

1. Line 168, "a 54% April decrease" should be rephrased, for example"54% decrease in April".

   Response: We made the improvement as suggested.

2. Line 197, figure 5c should be changed in 5d.

   Response: For the "(Figure 5c-f)" reference, we decided to keep 5c because it shows historic values to compare with.

3. Figure 7, d is never mentioned in text.

   Response: We included a reference for 7d in the text at page 14, line 236 of the preprint:

   "*Changes to the amount of annual ROS melt and frequency of ROS events are not correlated with historic winter and spring total precipitation amounts (Figure 7c and d), as the type of precipitation is more influential, and depends on air temperatures (and thus latitude).*"

4. In figure 3 add in title of figure 3 b, d and f that it is for the period of the mid-century for clarity.

   Response: Unfortunately, we could not fit the extra title in the figure space, but we kept the reference to mid-21$^{st}$ century in the caption.

5. In figure 4 some alterations on the axis titles can clarify the graphs. In 4b "snowmelt" can be clarified by writing "total snowmelt" and for c "proportion ROS" can be "proportion melt by ROS".

   Response: We updated the figure as suggested.

References

Jeong, D. Il and Sushama, L.: Rain-on-snow events over North America based on two Canadian regional climate models, Clim. Dyn., 50, 303–316, https://doi.org/10.1007/s00382-017-3609-x, 2018

---

## Author Response (AR1)

**Reviewer 1**
General comments:

This study is a novel investigation that is of interest to the professional community and in-line with the aims and scope of the journal. The topic is appropriately introduced with justification provided for the specific objectives. While some additional details on the statistical testing could be added (see below), the methodological approach appears logical and reproduceable. The results are organized around specific themes with figures that enhance understanding and are aligned with the final conclusions. Prior to supporting acceptance and publication, there are a small number of outstanding concerns with the manuscript that are addressed below as specific comments.

> Response: We thank the reviewer for the thorough and helpful review that has improved the quality of the manuscript.

Specific comments:

The proportions of historical ROS melt [to total melt] is larger here than a variety of previous findings for the region. For instance, Welty and Zeng (2021) find extreme ROS occurrence is approximately 24% for the Great Lakes basin, similar to the value the authors give on line 34 at over 25% of extreme ablation events being ROS. Looking at all ROS events, not just extreme, the maximum value to date I am aware of for this region is found in Suriano (2022). This notes between 30-50% of ablation is ROS in the eastern lakes, compared to less than 20% in the extreme northern/western regions. While the results here have a similar spatial pattern to Suriano (2022), with more ROS in the eastern lakes and less to the north and west, the magnitudes are rather different. Given one of the primary results of this study is the detection of large decreases in ROS events under the RCP4.5 scenario relative to historical period, it is warranted to provide further discussion on the robustness of the historical model values relative to observations. This appears absent from the manuscript currently and should be incorporated into the discussion section of the revision.

> Response: For our study, we used the definition of Jeong and Sushama (2018) to define an ROS event, as this definition was being used by them to project future climate impacts using RCP's across North America, and was based on the ROS definitions of studies before them. Thus, we defined an ROS event as >1 mm rainfall on >1 mm SWE and snowmelt occurring, so our results would be directly comparable with theirs. We now include an additional Discussion section (4.2) and a figure that discusses the comparability of our findings of historic ROS melt with other studies, and objectively evaluates our model results against historic observed data.
>
> *"Previous work by Jeong and Sushama (2018), whose definition of ROS we adopted, produced estimates of historic frequencies of ROS events comparable to ours:*

*approximately 10-20 ROS days per year in the Great Lakes Basin. Also, Jeong and Sushama (2018) report a 1976-2005 average annual amount of ROS runoff of approximately 100 mm or greater throughout the Basin, which is similar to our historic (1960-1999) estimates that were approximately 75 mm annual ROS melt in the southwest part of the Basin and 175 mm in the northeast. Jeong and Sushama (2018) evaluated their model results using observations and found that spatial patterns in ROS were captured reasonably well, although some errors likely were due to data uncertainties rather than model errors. Using a different definition of an ROS event (air temperature >0 °C and precipitation >5 mm during 2-day extreme snowmelt events), Welty and Zeng (2021) produced far fewer ROS events than we did. Additionally, Suriano (2022) defined an ROS event as a snow depth decrease of at least 1 cm with average daily temperature >0 °C, at least 0.01 cm precipitation, and no more than 2.54 cm snowfall (by depth) the previous day, over a 1960-2009 historic period. With this definition, Suriano (2022) reported a historic frequency of approximately 5 to 15 ROS events per year in the Great Lakes Basin, which is slightly less than our estimate of approximately 10-20 annual ROS events during 1960-1999.*

*To verify our historic estimates, we identified ROS amounts and frequencies in observed data using the same approach and definition as our GCM-forced SWAT model. The historic climate observations were from Maurer et al. (2007), used in Myers et al. (2021b), and our historic SWE observations were from Myers et al. (2021b), which had been estimated from the daily gridded North American snow depth dataset (Mote et al., 2018), both using the same 1° latitude/longitude grid with 50 evaluation points over the Great Lakes Basin. We found that for historic annual estimates of ROS melt, the mean among the gridded evaluation points for our GCM ensemble was 120 mm, while the mean calculated from observations was 118 mm, which was not a significant difference (two-sample t-test, p=0.90). For individual evaluation points, the estimates of annual ROS melt were positively related and had an MAE of 33 mm (Fig. S2a). This suggests that our GCM ensemble produced reasonable estimates of historic ROS melt amounts in the Basin. We also found that the historic observations produced an average annual ROS frequency of 20 days across the evaluation points, which was greater than the mean of 12 days estimated by our GCM ensemble for the points over our historic 1960-1999 period (p<0.001). This was because our ROS definition included historic observed events that were the result of natural stochasticity in snowpack SWE amounts (i.e., sporadic daily increases or decreases in the SWE data due to factors such as the timing of measurements at different stations, rather than "clean" modeled melt; Suriano and Leathers, 2017; Suriano, 2022). Thus, our definition underestimated the frequency of ROS days when applied to model data, due to the lack of additional stochastic small melt events identified by the criteria, producing an MAE of 8 days between observations and modeled ROS frequency (Fig. S2b). However, when ROS amounts are accumulated over the season, this issue is remedied (Fig. S2a)."* (page 20, line 362 to page 21, line 391)

[Figure]

**Figure S2.** For the 50 gridded climate and snowpack evaluation points in the Great Lakes Basin: a) comparison of historic (1960-1999) mean annual ROS melt amounts calculated for observed data with those modeled by our ensemble of climate projections, and b) the same comparison for the mean annual frequency of ROS events.

References:

Jeong, D. Il and Sushama, L.: Rain-on-snow events over North America based on two Canadian regional climate models, Clim. Dyn., 50, 303–316, https://doi.org/10.1007/s00382-017-3609-x, 2018

Suriano, Z. J.: North American rain-on-snow ablation climatology, Clim. Res., 87, 133–145, https://doi.org/10.3354/CR01687, 2022.

Suriano, Zachary J., and Daniel J. Leathers. Spatio-temporal variability of Great Lakes basin snow cover ablation events. Hydrological Processes 31.23, 4229-4237, 2017.

Welty, Josh, and Xubin Zeng. Characteristics and causes of extreme snowmelt over the conterminous United States. Bulletin of the American Meteorological Society 102.8, E1526-E1542, 2021.

The authors acknowledge on line 126 the threshold used for statistical significance for their correlation tests. However, it is unclear if any significance testing was conducted for the rest of the study. Was any sort of t-test or difference of means testing conducted for the results comparing the historical period to the mid-century period? If not, this should be considered by the authors to aid in differentiating meaningful changes from ones still within the noise.

Response: We now have included significance testing for our comparisons of ROS and climate between the historic and mid-21st century periods. We also now describe the approach for this in the methods, and provide instances where significance testing is used below. However, throughout our revision we keep effect size as the focus, rather than statistical significance, following the guidance of previous work (Wasserstein et al., 2019; Ziliak & McCloskey, 2008)

"*For comparisons between time periods, statistical significance was evaluated by comparing annual area-weighted ensemble-average values for the Great Lakes Basin between the historic (1960-1999, n=40 years) and mid-21st century (2040-2069, n=30 years) periods using two-tailed unpaired t-tests.*" (page 6, lines 138-141)

"*Spatially averaged annual precipitation increases by 53 mm (6.3%) from 839±63 mm (mean and standard deviation of GCM ensemble) during 1960-1999 to 892±77 mm by the mid-21st century (p<0.001), while spatially averaged annual air temperatures increase by 2.7 °C from 5.2±0.7 °C during the 1960-1999 period to 7.9±1.0 °C (p<0.001).*" (page 7, lines 146-149)

"*Further, our model shows that winter+spring rain to snow ratios over the basin (calculated by dividing the total winter+spring rainfall by total winter+spring snowfall) increase from around 1.5 historically to 1.9 by mid-century (p<0.001), which means that proportionally more rainfall could contribute to the declines in snowmelt and snowpack SWE.*" (page 7, lines 155-158)

"*Overall, at the major river basin scale using the RCP 4.5 scenario, the ensemble average amount of annual snowmelt during ROS events changes by -42% to +1%, with a basinwide area-weighted average of -22% (p<0.001).*" (page 13, lines 211-213)

References:

Wasserstein, R. L., Schirm, A. L., & Lazar, N. A. (2019). Moving to a world beyond "p< 0.05". The American Statistician, 73(sup1), 1-19.

Ziliak, S., & McCloskey, D. N. (2008). The cult of statistical significance: How the standard error costs us jobs, justice, and lives. University of Michigan Press.

**Reviewer 2**
A representative simulation of ROS melt events is important for improving hydrological modeling practice in snow dominated region. It is valuable to look into the future impact of ROS melt events under climate change. This is exactly what this work intends to address. However, the current manuscript is not yet ready for publication, due to two points :

Response: We thank the reviewer for the thoughtful feedback which we have used to improve the quality and clarity of the manuscript.

1. This work utilized a calibrated SWAT ROS model to simulate the hydrological process using CMIP5 climate projections. All analyses are based on the assumption that this calibrated model is representative. However, as described "The SWAT ROS model for the Great Lakes Basin simulated historic streamflow at the daily time step with an NSE of 0.38 (with 29% of stations greater than 0.5) and a dr of 0.62 (Myers et al., 2021b). The model simulated historic snowpack SWE at the daily time step with an MAE of 26 mm", the model cannot be well considered well-calibrated with a low NSE of 0.38 for discharge simulation. Moreover, 26 mm MAE for daily SWE is a considerable high bias in comparison to the SWE value of the study area (e.g. Figure 4). The median SWE value of many months is around 50 mm or lower. GCM climate projections are highly uncertain already. A hydrological model with high bias will make the combination much worse. As a consequence, it is not reasonable to trust the analyses of this work about future climate change impact, even the analysis strategy is comprehensive. Therefore, the authors should implement the climate change investigation based on a reasonably well-calibrated SWAT ROS model. Moreover, detailed information about the rationality of the calibrated SWAT model is necessary but missing. Such information should be properly added to this paper or its supplementary material for its readers. The authors simply cited the paper that developed and evaluated the SWAT ROS model (reference below). But it is not open-access.

Myers, D. T., Ficklin, D. L., and Robeson, S. M.: Incorporating rain-on-snow into the SWAT model results in more accurate simulations of hydrologic extremes, J. Hydrol., 603, 126972, https://doi.org/10.1016/J.JHYDROL.2021.126972, 2021b.

Response: We attempted several new calibrations of the Great Lakes Basin SWAT ROS model aimed to improve model performance, including expanding parameter ranges within physical reason and removing the least sensitive parameters from the calibration. However, we were unable to improve model performance for simulating streamflow and snowpack beyond that of the Myers et al. 2021b model, which had been heavily experimented on with different calibration strategies and evaluations during that study. We believe that the accuracy of our model is acceptable considering the error of all the spatially aggregated datasets that go into it (climate, snowpack, soils, etc.), and especially when considering the use of our daily time step, which incorporates high temporal scrutiny in our evaluations over a large geographic area. For instance, the error for the snow depth data that we modeled SWE from and evaluated our model against is expected to average 1.0 to 2.5 cm depth in the Great Lakes Basin (Kluver et al., 2016; Mote et al., 2018). Uncertainties in gridded climate forcings could also affect performance of hydrological outputs (Maurer et al., 2010; Muche et al., 2020; Stern et al., 2022).

We have expanded our description of model evaluations to include more synopses from the Myers et al. 2021b study and a figure that depicts the locations of stations and their performance. This study optimized our parameter set to the best average performance of 99 streamflow and 50 snowpack calibration stations across the Basin. Figure 2 (below) now shows how stations that perform well for streamflow and snowpack simulation are dispersed throughout the Great Lakes Basin. We also reference literature about satisfactory model performance interpretations at small temporal scales (Kalin et al., 2010). Although the Myers et al. 2021b paper is not open access, we are happy to share this paper, through the editor if you prefer. We provide the updated text below.

*"The SWAT ROS model for the Great Lakes Basin simulated historic streamflow at the daily time step with an average NSE of 0.38 (with 29% of stations greater than 0.5, 48% greater than 0.4, and a maximum NSE of 0.71) and an average $d_r$ of 0.62 (Myers et al., 2021b). The model simulated historic snowpack SWE at the daily time step with an MAE of 26 mm (Figure 2a-c). Previous work by Kalin et al. (2010) has stated that arbitrary interpretations of performance metrics for models at small temporal scales should be relaxed compared to what would be expected for models at coarse (e.g., monthly) time steps, for instance that an NSE between 0.3 and 0.5 could fit criteria for satisfactory model performance in some contexts. Calibrated parameters for this model can be found in Table S1 in Myers et al. (2021b). We also investigated seasonal model performance for simulating snowmelt (water equivalent) during only days when ROS melt was occurring, and found that the SWAT ROS model we use had an MAE of 8.6 mm, 9.4 mm, and 5.8 mm for simulating snowmelt on those days in the winter, spring, and fall, respectively."* (page 4, lines 84-93)

[Figure]

**Figure 2.** Evaluation statistics for simulating historic a) streamflow using Nash Sutcliffe Efficiency (NSE), b) streamflow using revised Index of Agreement ($d_r$), and c) snowpack using mean absolute error (MAE) at the daily time step. Adapted from Myers et al., (2021b).

In response to the reviewer's other comments, we added a section (4.2) that discusses our historic ROS estimates in comparison with other studies (also described in our public comment to Reviewer 1). In this section, we evaluate the historic ROS melt amount and ROS frequency estimates of our GCM ensemble against estimates from historic observations. This showed that our SWAT ROS model with GCM forcings was

reasonably simulating ROS melt in the Great Lakes Basin in comparison with historic data.

Kalin, Latif, et al. "Predicting water quality in unmonitored watersheds using artificial neural networks." Journal of Environmental Quality 39.4 (2010): 1429-1440.

Kluver, D., Mote, T., Leathers, D., Henderson, G. R., Chan, W., & Robinson, D. A. (2016). Creation and validation of a comprehensive 1° by 1° daily gridded North American dataset for 1900-2009: Snowfall. Journal of Atmospheric and Oceanic Technology, 33, 857–871. https://doi.org/10.1175/JTECH-D-15-0027.1

Maurer, E. P., Hidalgo, H. G., Das, T., Dettinger, M. D., & Cayan, D. R. (2010). The utility of daily large-scale climate data in the assessment of climate change impacts on daily streamflow in California. Hydrology and Earth System Sciences, 14(6), 1125-1138.

Mote, T. L., Estilow, T. W., Henderson, G. R., Leathers, D. J., Robinson, D. A., & Suriano, Z. J. (2018). Daily gridded north American snow, temperature, and precipitation, 1959-2009, version 1. Boulder, Colorado USA: NSIDC: National Snow and Ice Data Center. N5028PQ3. Https://Nsidc.Org/Data/G10021/Versions/1.

Muche, M. E., Sinnathamby, S., Parmar, R., Knightes, C. D., Johnston, J. M., Wolfe, K., ... & Smith, D. (2020). Comparison and evaluation of gridded precipitation datasets in a Kansas agricultural watershed using SWAT. JAWRA Journal of the American Water Resources Association, 56(3), 486-506.

Stern, M. A., Flint, L. E., Flint, A. L., Boynton, R. M., Stewart, J. A. E., Wright, J. W., & Thorne, J. H. (2022). Selecting the optimal fine-scale historical climate data for assessing current and future hydrological conditions. Journal of Hydrometeorology, 23(3), 293-308.

2. Future climate projects have large uncertainty. When evaluating climate change impacts, it is more reasonable to discuss the trend or relative changes rather than absolute quantities. The authors should shorten such contents and keep the necessary ones only. Besides, Figure 2 shows different behaviors of climate driving force during different future periods. It would be interesting to investigate the corresponding hydrological signatures of different future periods. Although, as described in section 2.4, the analyses of future period include mid-21st century and late-21st century. Throughout the paper, the result and analysis of late-21st century is almost none. Please complete such missing parts.

Response: We now primarily focus on relative changes in our analyses, while some absolute changes are also mentioned to provide context for the magnitude of the changes. An example is below.

"*For instance, the area-weighted median (value of a ranked set where half the total area is ranked lower; Willmott et al., 2007) amount of January ROS melt among subbasins increases 59% from 3.2 mm historically to 5.1 mm by mid-21st century (Figure 5a). Similarly, area-weighted median February ROS melt rises 50% from 7.0 mm historically to 10.5 mm in the mid-21st century. In the spring, the amount of ROS melt decreases due to the reduction in snowpack.*" (page 9, line 170 to page 10, line 174)

For consistency and clarity in our findings, we decided to remove the isolated references to late-21st century from the paper, and focus on mid-21st century changes only. We focus on mid-21st century because it is the period that the models (and RCPs) generally agree about climate changes, and because we expect that it will be the most meaningful focus for water resource managers in the Great Lakes Basin. Thus, we updated the figure to end after mid-21st century and clarified our text to alleviate any confusion about future time period.

[Figure]

**Figure 3.** Basinwide ensemble-average a) annual total precipitation, b) annual air temperature, c) winter+spring rain to snow ratio, d) winter+spring rainfall, and e) winter+spring snowfall from the climate input data, with 10-year averages (red lines), based on the RCP 4.5 pathway. Shading indicates historic 1960-1999 (red) and mid-21st century 2040-2069 (blue) periods, as well as ensemble standard deviations (grey).

**Community Comment 1**

This review was prepared as part of graduate program Earth & Environment (course Integrated Topics in Earth & Environment) at Wageningen University, and has been produced under supervision of dr Ryan Teuling. The review has been posted because of its potential usefulness to the authors and editor. Although it has the format of a regular review as was requested by the course, this review was not solicited by the journal, and should be seen as a regular comment. We leave it up to the author's and editor which points will be addressed.

Title paper: "Hydrologic implications of projected changes in rain-on-snow melt for Great Lakes Basin watersheds"

Overall impression

Rain on snow (ROS) melt events can have a big influence on their surroundings and can be either big or small. This manuscript investigates the impact of the climate change on the ROS events in the Great Lakes Basin in northern United States and few states of Canada for the period of 1960 to 2099, focussing on the period of 2040-2069 by the use of the model Soil and Water Assessment Tool (SWAT) with the addition of an energy budget equation to project the ROS events. With the results they looked at relationships and correlations between the different obtained variables. In general, the ROS events tend to happen earlier in the year by mid-21st century compared to the historical 1960-1999 values. The rain to snow ratio changes from around 1.5 historically to 2.0 at the end of the 21st century. This all also has influence on the water yield of the basins which also shift to earlier in the year.

The paper is nicely written and is important in these times of changing climate. The text has a good structure as the results are divided in understandable blocks. The variables are also captured in some good figures, although some changes should be made. Based on these comments, I would recommend publication with minor revisions.

> Response: We thank you, members of the Integrated Topics in Earth & Environment course, for the valuable feedback. We have incorporated the comments into our manuscript and present our responses. We also include you in the acknowledgements of our revision.

General comments,

Firstly, as definition of a rain-on-snow event in the analysis section is stated that an event occurs on days with >1 mm rainfall on >1 mm snowpack SWE. By reading the Jeong and Sushama,

2018 paper this definition is not complete. It should be days with >1 mm rainfall on >1 mm snowpack SWE and decreasing SWE. By using the wrong definition rain-on-snow events could be wrongfully depicted in the results, and thus possible differences in conclusions. This should be solved by correctly using the definition and changing this in the paper.

> Response: Thank you for pointing out our error in stating how we defined ROS events. You are correct that the definition must include snowmelt occurring, which we included in our script but hadn't mentioned in the text. We have now updated it to:
>
> "*ROS melt events were defined as days with >1 mm rainfall on >1 mm snowpack SWE and snowmelt occurring, which is a definition that has been previously used to model ROS events and project climate change impacts (Jeong and Sushama, 2018).*" (page 6, lines 133-135)

Secondly, in the research question is stated that the change of rain-on-snow melt and hydrology due to climate change will be assessed for the 21$^{st}$ century. In the method an argument is made about that for informing water resources management and because of better agreement with the models the primary focus will be on the mid-21$^{st}$ century (2040-2069). In the rest of the paper late-21$^{st}$ century is only mentioned for the change in ratio of area-weighted winter+spring rain-to-snow. Which has only a 0.1 change to the ratio change of the mid-21$^{st}$ century where the mid-21$^{st}$ century has a better agreement to the models and thus will probably be more accurate. Also in the conclusion at the end is stated "could help prepare ….. for the climatic changes of the 21$^{st}$ century and beyond" where nothing is known for this latter time period with results from this paper. Thus, either the research question has to be rewritten to only the mid-21$^{st}$ century together with the conclusion or the late-21$^{st}$ century should be included in the rest of the analysis.

> Response: Thank you for pointing out this need for consistency. We have removed references beyond the mid-21$^{st}$ century from the research question and conclusions, as well as other locations where late-21$^{st}$ century had been mentioned in the text, to focus our study consistently on mid-21$^{st}$ century changes.
>
> "*Our research asks, "How does ongoing climate change alter ROS melt and hydrology in the Great Lakes Basin by the mid-21st century?"*" (page 2, lines 44-45)
>
> "*The implications of this work, specifically involving the influence of changing ROS melt on extreme hydrological events and future water availability, as well as the climate-related sensitivities to changing ROS melt, could help prepare managers of ecosystems and human water uses for the climatic changes of the mid-21st century.*" (page 22, lines 422-425)

Lastly, again a comment on the research question but now because change in hydrology due to climate change is asked in the question. When stating hydrology I expect more variables to be analysed then only water yield, the rest of the analysed variables either belong to the change in climatic values such as rain and temperature or change in rain-on-snow events. Also, in the methods groundwater is mentioned to be modelled (line 65) but later not analysed in the results. So, either the phrasing of the research question should be altered or more hydrological variables should be assessed in the paper. This is important as the goal of this paper is to inform water managements to prepare for the changes due to climate change. For them the groundwater or runoff variables are also very important, and as they change with changing ROS melt (as said in line 39-41) this should be addressed.

> Response: Our results focused on water yield because it is normalized streamflow, which our model was calibrated and evaluated for. Similarly, we report snowpack that our model was calibrated and evaluated for. Other variables such as groundwater and soil water clearly are important, but we provided limited results for them, because of space limitations or because they had not been specifically calibrated and evaluated for using the observed data. This is definitely an area of future research that we hope to look into further. Thus, we added a sentence about the importance of this additional research.
>
> "*Future work could also investigate how changing ROS conditions affect other components of the water balance including groundwater and soil water storage in the Great Lakes Basin.*" (page 22, lines 421-422)

Comments,

1. In lines 312 -320, Myers mentions the difference in findings with Surianon mentioning that the studied times differ. But as the simulation used in this study is done for 1960-2099, the same time periods could be compared as the data will be present after simulations. Why not analyse the same time period (1960-2009) as Suriano to make this comparison possible, to diminish this suggestive difference.

> Response: We decided to remove this comparison since it was not definitive, and we now provide more concrete details about historic ROS in a new Discussion section 4.2 (described in our public comment to Reviewer 1). We used 1960-1999 as the historic time period for this study because that matches previous work we have performed and evaluated (Myers et al. 2021b) using our SWAT ROS model, and fits with the time period of the historical climate observations we used (Maurer et al. 2007).
>
> Maurer, E. P., Brekke, L., Pruitt, T., and Duffy, P. B.: Fine-resolution climate projections enhance regional climate change impact studies, Eos, Trans. Am. Geophys. Union, 88, 504, https://doi.org/10.1029/2007eo470006, 2007.

Myers, D. T., Ficklin, D. L., and Robeson, S. M.: Incorporating rain-on-snow into the SWAT model results in more accurate simulations of hydrologic extremes, J. Hydrol., 603, 126972, https://doi.org/10.1016/J.JHYDROL.2021.126972, 2021b.

2. Lines 338-340, the speculation of influence of rain-to-snow ratio on size and timing of spring snowmelt and summer baseflow is made. This could be analysed by simply calculating the correlation between the COV (center of volume) and the rain-to-snow ratio which are variables present in the results.

    Response: We updated the Discussion text with our results for this relationship. We decided instead of the correlation to state the size of basin-average effects, which could be even more applicable to water resources management.

    "*Thus, the rain to snow ratio could help explain the earlier COV of ROS melt for the Great Lakes Basin by mid-21st century, since we found that as the basin-average rain to snow ratio increases from approximately 1.5 historically to 1.9 by mid-21st century, the COV of ROS melt occurs two weeks earlier.*" (page 20, lines 347-349)

3. Line 99, "thus. Nineteen climate models ...... were used". There is a missing argument about why you use 19 models instead of more or less. Please add an argument.

    Response: To improve clarity in this choice, we added the following:

    "*We chose to include nineteen climate models because that was the total number of models using the RCP 4.5 scenario that had been downscaled and bias corrected in the multi-model ensemble (Maurer et al., 2007).*" (page 5, lines 111-113)

    Maurer, E. P., Brekke, L., Pruitt, T., and Duffy, P. B.: Fine-resolution climate projections enhance regional climate change impact studies, Eos, Trans. Am. Geophys. Union, 88, 504, https://doi.org/10.1029/2007eo470006, 2007.

4. Line 165, the word "drastic" is used. Drastic is not a quantitative value as it is more an subjective use when not supported with arguments. Either rewrite the sentence or add an argument.

    Response: We removed the word to avoid the subjectivity.

5. Lines 103-105, this information is a good/ better argument for the statement in line 118-119. As it now seems as random added information but potentially better used in the latter argument.

   Response: We moved the text as suggested.

6. Line 167 , it is as a reader unclear whether the value mentioned for March is also the maximum or the value at the time of the maximum of April. This should be clarified.

   Response: We reworded this sentence for clarity:

   "*Historically the maximum snowmelt overall has been in April with an area-weighted median of 85.3 mm, while the March median snowmelt has been less at 44.8 mm. By mid-21st century the median amount of monthly snowmelt among subbasins reaches a maximum at 44.8 mm in March, but drops to only 39.5 mm in April, which is a 54% decrease during April between the two periods*." (page 10, lines 179-182)

7. Line 172, it is unclear if "proportion of melt" means temperature based melt or ROS based melt. This should be clarified.

   Response: We added an explanation to be clearer:

   "*However, the proportion of melt occurring during December ROS days (compared with all December melt) decreases from an area-weighted median of 71% historically (1960-1999) to 59% by mid-21st century (a decrease of 12%)*." (page 10, lines 186-188)

8. Comment on figures in general: in the captions some abbreviations are written out where others are only posted as abbreviation. For the consistency of the paper this should be the same in all figures.

   Response: We fixed the figure captions so that the abbreviations are described at the first mention, and then abbreviated each time after.

9. Line 212, why is 2050s mentioned here instead of the same formulation used in the rest of the paper: "mid-21st century"? This could confuse the reader.

   Response: We changed it to "mid-21st century" to avoid the confusion.

10. Figure 6, Instead of "high flows" state "high water yields" in figure title of 6 e and f as stated in the caption for clarification.

Response: We updated the figure as suggested, but for space wrote "high WYLD" in the figure title then defined WYLD as water yield in the caption.

11. Figure 7, could be better depicted when the figures with frequencies (b, d and f) are on the right and with melt (a, c and e) on the left.

Response: We updated the figure as suggested.

Specific comments,

1. Line 168, "a 54% April decrease" should be rephrased, for example"54% decrease in April".

Response: We made the improvement as suggested.

2. Line 197, figure 5c should be changed in 5d.

Response: For the "(Figure 5c-f)" reference, we decided to keep 5c because it shows historic values to compare with.

3. Figure 7, d is never mentioned in text.

Response: We included a reference for 7d in the text:

"*Changes to the amount of annual ROS melt and frequency of ROS events are not correlated with historic winter and spring total precipitation amounts (Figure 7c and d), as the type of precipitation is more influential, and depends on air temperatures (and thus latitude).*" (page 15, lines 248-250)

4. In figure 3 add in title of figure 3 b, d and f that it is for the period of the mid-century for clarity.

Response: We made the improvement as suggested (now labelled Figure 4 in the revision).

5. In figure 4 some alterations on the axis titles can clarify the graphs. In 4b "snowmelt" can be clarified by writing "total snowmelt" and for c "proportion ROS" can be "proportion melt by ROS".

Response: We updated the figure as suggested.

References

Jeong, D. Il and Sushama, L.: Rain-on-snow events over North America based on two Canadian regional climate models, Clim. Dyn., 50, 303–316, https://doi.org/10.1007/s00382-017-3609-x, 2018

---

## Author Response (AR2)

Dear authors,

thank you for sumbitting your revised manuscript and responses to the reviewer reports. While all comments by referee #1 have been cleared, the report of referee #2 indicates that there is still need for clarification, especially regarding important aspects of model performance and scope. I invite you to respond to these comments and provide a manuscript with corresponding revisions.

Kind regards,
Daniel Viviroli

> Response: We thank the editor for all the help with the review and suggestions for improving the manuscript. Below are our responses to the reviewer comments.

**Reviewer 1**

> Response: We thank the reviewer for the helpful comments on the manuscript and the positive feedback.

**Reviewer 2**

The manuscript has been improved according to the comments in round 1. Especially, the climate projection sections read more reasonable now. But (there is a but), the model rationationty part, as indicated in my first comment of last round, is still not convincing. This is the basis for whatever outcomes of the future projection investigations in this work. Because ROS is such an interesting research topic and this work could be beneficial for the community, I dig into the methodology paper (Myers et al., 2021b) that this manuscript is based on. To make it clear, let me zoom into the specific points:

> Response: We thank the reviewer for providing helpful comments to improve those sections, and for providing constructive feedback for what still needs improvement. We hope our responses can satisfactorily address the remaining concerns.

1. Low average NSE for discharge simulation. "The SWAT ROS model for the Great Lakes Basin simulated historic streamflow at the daily time step with an average NSE of 0.38 (with 29% of stations greater than 0.5, 48% greater than 0.4, and a maximum NSE of 0.71)". It is not acceptable, in particular, the author wants to address the effects on extreme high water yield (as defined by the authors: "Finally, the SWAT model outputs for water yield represent the area-averaged water export through the outlet in mm."). with such a definition, streamflow is more or less equivalent to or contributes a lot to the water yield of this work. Mathematically, NSE is highly affected by peak flow which is highly relevant to extreme high water yield. Low NSE

largely indicates a bad fitting of the peaks. Thus, it is not reasonable to discuss the implications on extreme water yield, if the model cannot represent them properly.

Response: Following the reviewer's feedback and further research, we removed the analysis regarding extreme winter and spring water yields from the results, as well as its mention in the methods and conclusions sections. We now introduce it as a potential avenue for further research. We also elaborated on how the model performance guides the scope of investigations to only those most reliable and representative.

[revised manuscript text omitted]

Thoughts: After checking the methodology paper (Myers et al., 2021b), it seems that the model simulated somehow descent streamflow in winter/spring period but performed bad in summer period. If ROS is more important for winter/spring period (I am not a snow scientist. This statement should be double checked), perhaps calculating the metrics and proving the reasonable representation of streamflow in these seasons is a breakthrough point.

Response: We appreciate the reviewer for the constructive and helpful idea. We explored the impact of using only winter and spring data in our evaluations. However, the evaluation statistics were similar to before, with a streamflow NSE of 0.35, streamflow $d_r$ of 0.64, and snowpack MAE of 37 mm. Thus, we decided to stick with the evaluations as reported in the Myers et al., 2021b study.

Minor suggestion: higher temporal resolution does not necessarily mean using a lower criterion for evaluating the model performance. And the spatial side should not be ignored. In this work, the spatial resolution is coarse. I suggest to remove the statement "for instance that an NSE between 0.3 and 0.5 could fit criteria for satisfactory model performance in some contexts".

Response: We removed the statement as suggested.

2. High MAE for daily SWE. The authors added some descriptions of snow melt performance as supporting references of reasonable snow simulations. However, acceptable MAE snowmelt simulations do not mean reasonable performance of SWE outputs. It may indicate the melt module works, and something has to be improved in the snow accumulation module. The authors should clarify this point: is 26 mm MAE for daily SWE is an acceptable bias in the study region or not? Particularly, as shown in Figure 5, the median SWE value of many winter months is around 50 mm or lower. As a reader, I will interpreter 26 mm MAE as a large error.

Response: We now add more clarification about our snowpack SWE errors and how they were distributed across the Basin, with larger absolute errors in stations that had larger historic snowpack amounts, and errors less than the mean of 26 mm MAE in stations with smaller historic snowpack amounts.

"*Daily snowpack SWE error across the Basin ranged from <20 mm MAE throughout the southwest subbasins to approximately 40-70 mm MAE in the northeast (Figure 2c). This spatial variation in MAE scaled with the average observed daily snowpack SWE during winter and spring across the Basin, which ranged from approximately 50-100 mm in the southwest subbasins to over 150 mm in the northeast, described in Section 3.4. Thus, subbasins with lower amounts of observed SWE would also have smaller errors than the average of 26 mm. We find this measure of absolute error to be acceptable, particularly considering the errors inherent to the gridded snowpack data we compare against (e.g., spatial averaging and simplification of accumulation and ablation processes during conversion from snow depth; Myers et al., 2021b; Ensor and Robeson, 2008; Hill et al., 2019). Mean absolute errors in modeling the gridded snowpack SWE have been found to vary temporally as well, for instance being 12.7 mm MAE on January 1, 45.1 mm MAE on February 1, 26.8 mm MAE on March 1, and 9.6 mm MAE for April 1, 1978, across the Great Lakes Basin in comparison with station measurements, and proportional to the amount of snowpack on the ground during those days (Myers et al., 2021b).*" (page 4, line 101 to page 5, line 112)